# Global Directional Priors with Local Statistical Validation for Scalable Causal Discovery

Wei Yuan [1]    Zixuan Shao [2]    Shuhui Wang [3]

## Abstract

Constraint-based causal discovery relies on conditional independence (CI) tests whose reliability degrades as conditioning sets grow, particularly in hub-dominated graphs. Existing methods constrain adjacency or global structure, but leave conditioning-set dimensionality uncontrolled. In this paper, we propose Ordering-Constrained Markov Blanket discovery (OCMB), a paradigm that treats conditioning-set dimensionality as a first-class constraint. OCMB decouples discovery into two stages: lightweight global ordering estimation providing directional priors, followed by local Markov blanket validation within small, ordering-constrained candidate sets. By enforcing directional constraints before any CI test, OCMB ensures bounded conditioning sets even with hub nodes. We show that OCMB recovers correct parent sets provided a high-recall ordering assumption holds, without requiring the ordering to be globally correct. Experiments demonstrate that OCMB significantly improves precision and robustness over constraint-based and hybrid methods in high-dimensional regimes where conventional CI-based approaches fail.

## 1. Introduction

The problem of causal discovery from observational data is fundamental in machine learning, with applications spanning scientific discovery and decision-making. Constraint-based methods such as PC (Spirtes et al., 2000) remain attractive due to their statistical guarantees, but face scalability issues in high-dimensional settings, particularly in hub-dominated graphs where CI tests require large conditioning sets.

Existing methods primarily differ in how they constrain the search space. Constraint-based approaches restrict adjacency via skeleton discovery; score-based methods constrain structures via acyclicity-aware optimization (Chickering, 2002; Zheng et al., 2018); hybrid methods like MMHC (Tsamardinos et al., 2006) first discover undirected skeletons then orient edges. However, these paradigms assume CI tests remain reliable as conditioning sets grow. In hub-dominated graphs, this assumption fails: conditioning-set sizes quickly become prohibitive, causing statistical errors to cascade.

In this work, we argue that conditioning-set dimensionality should be a first-class constraint in causal discovery. Rather than constraining adjacency or structure directly, we constrain the directional search space globally, ensuring all subsequent CI tests operate under bounded conditioning sets.

We propose Ordering-Constrained Markov Blanket discovery (OCMB), a two-stage paradigm decoupling global directionality estimation from local structural validation. Stage I infers a global ordering providing directional priors that restrict candidate parent sets; Stage II performs CI tests only within small, ordering-constrained candidate sets (Figure 1). The ordering acts as a soft directional filter rather than a hard structural constraint. Unlike MMHC, which performs CI tests on high-dimensional neighborhoods during skeleton discovery (Figure 2), OCMB enforces directional constraints before any CI test. This shift from undirected skeleton-first to global direction-first maintains statistical reliability in hub-dominated graphs. We instantiate Stage I with CaPS (Xu et al., 2024); other backbones (DiffAN (Sanchez et al., 2023), SciNO (Kang et al., 2025)) are compatible.

Crucially, OCMB does not require the ordering backbone to recover the true causal ordering. The ordering serves as a screening mechanism to construct tractable candidate supersets. We provide theoretical guarantees showing that under a high-recall ordering assumption, where true parents rank ahead of most non-parents, OCMB correctly recovers

[1]University of the Chinese Academy of Sciences, Beijing, China [2]University of California, San Diego, La Jolla, CA, USA [3]State Key Lab of AI Safety, Institute of Computing Technology, Chinese Academy of Sciences, Beijing, China. Correspondence to: Shuhui Wang <wangshuhui@ict.ac.cn>.

*Proceedings of the 43rd International Conference on Machine Learning*, Seoul, South Korea. PMLR 306, 2026. Copyright 2026 by the author(s).

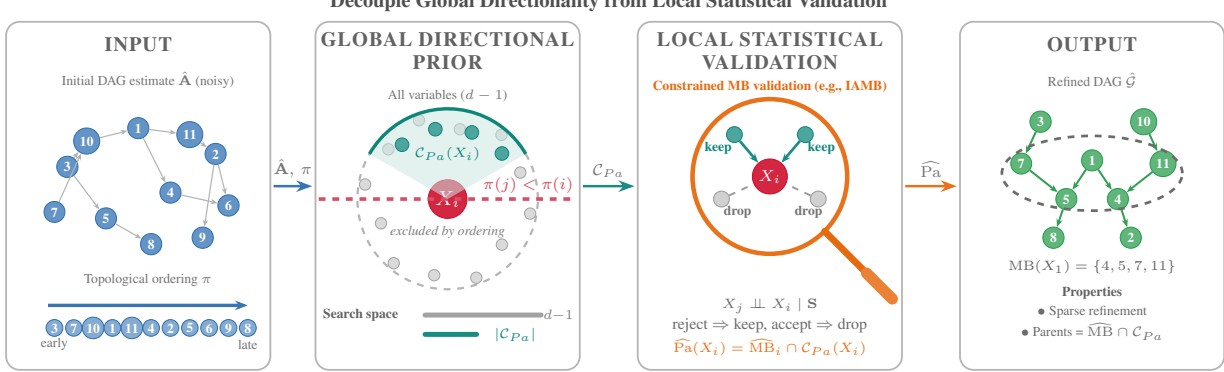

Figure 1. **OCMB Framework.** Stage I uses the ordering constraint to restrict parent candidates to predecessors $\{j : \pi(j) < \pi(i)\}$, reducing search space from $O(d)$ to $O(K)$. Stage II performs constrained Markov blanket validation and constructs edges only from $\widehat{\mathrm{Pa}}(X_i) = \widehat{\mathrm{MB}}_i \cap \mathcal{C}_{Pa}(X_i)$, yielding an ordering-consistent sparse DAG.

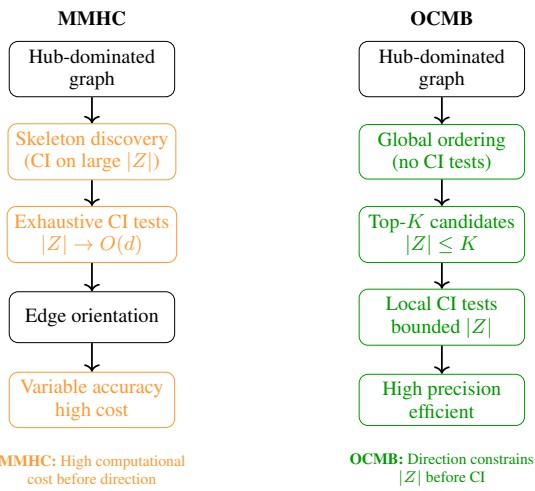

Figure 2. MMHC performs CI tests on large neighborhoods during skeleton discovery, where statistical reliability breaks down. OCMB enforces directional constraints before CI testing, ensuring bounded conditioning sets.

parent sets. This clarifies ordering's role as an inductive bias, not a definitive structural estimate.

Empirically, OCMB excels in hub-dominated graphs where classical constraint-based methods fail due to high-dimensional conditioning. Our experiments demonstrate that ordering constraints substantially reduce CI test dimensionality, leading to lower error rates and improved structural recovery.

**Conflict of Interest Disclosure.** The authors declare no financial conflicts of interest related to this work.

## 2. Related Work

*Constraint-based methods* (PC, FCI (Spirtes et al., 2000)) infer structure via CI testing but suffer from statistical unreliability around hub nodes where conditioning sets grow large. *Score-based methods* (GES (Chickering, 2002), NOTEARS (Zheng et al., 2018)) optimize scoring functions over DAGs; neural extensions (DAG-GNN, GraN-DAG) handle nonlinearity but face scalability challenges. *Ordering-based methods* (CaPS (Xu et al., 2024), DiffAN, SciNO (Kang et al., 2025)) estimate topological orderings via score matching, then threshold edge scores to obtain a DAG. CaPS (Xu et al., 2024) uses permutation-based score matching under additive noise models; SciNO (Kang et al., 2025) extends this to more general nonlinear settings. While ordering-based methods are efficient, orderings alone produce spurious edges without local validation. OCMB uses orderings as a *directional prior* ("backbone") to construct candidate sets for subsequent CI-based refinement.

*Bounded-conditioning-set methods* address high-dimensional CI testing by limiting conditioning-set sizes. Textor et al. (2015) and Wienöbst et al. (2020) show that under graphical constraints, CI tests with bounded conditioning sets can recover causal structures. Kocaoglu (2023) and Liu et al. (2024) extend these ideas to broader settings. These methods impose a *global cap on conditioning-set order*, whereas OCMB uses an ordering prior to restrict the *candidate search space before CI testing*. The two approaches are complementary: bounded-order CI methods constrain the conditioning dimension directly, while OCMB constrains the search space through directional priors, achieving bounded conditioning sets as a consequence.

*Markov blanket discovery* methods such as IAMB

(Tsamardinos et al., 2003b), HITON-MB (Aliferis et al., 2010), and MMPC (Tsamardinos et al., 2003a) identify the local neighborhood of each variable via iterative CI testing. These methods are well-suited for local structure learning but may require high-dimensional conditioning around hub nodes. OCMB addresses this by restricting the candidate search space to ordering-feasible predecessors before applying MB discovery, reducing conditioning-set sizes.

Hybrid methods combining paradigms have gained attention: MMHC (Tsamardinos et al., 2006) uses constraint-based skeleton discovery followed by score-based orientation; EEMBI (Dong & Gao, 2025) exploits Markov blanket intersections. OCMB differs by using orderings as soft directional constraints that bound conditioning-set dimensionality, improving CI test reliability while preserving local statistical validation.

## 3. Method: OCMB

We consider learning a DAG $\mathcal{G} = (\mathcal{V}, \mathcal{E})$ among variables $\mathcal{V} = \{X_1, \ldots, X_d\}$ from i.i.d. samples $\mathbf{X} \in \mathbb{R}^{n \times d}$, under standard faithfulness and causal sufficiency assumptions (Appendix A). Let $\mathrm{Pa}_{\mathcal{G}}(X_i)$, $\mathrm{Ch}_{\mathcal{G}}(X_i)$, $\mathrm{MB}_{\mathcal{G}}(X_i)$ denote parents, children, and Markov blanket of $X_i$. Methods like CaPS (Xu et al., 2024) and SciNO (Kang et al., 2025) produce a topological ordering $\pi$ and parent scores $S \in \mathbb{R}^{d \times d}$, then threshold to obtain edges; while efficient, this yields many false positives. OCMB instead uses $(\pi, S)$ as a directional prior constraining the search space for local CI-based inference.

### 3.1. Ordering as a Directional Prior

Unlike skeleton constraints (undirected adjacency), ordering constraints specify directional feasibility: which causal directions are admissible. This ensures automatic acyclicity and reduces conditioning dimensions around hub nodes.

**Directional Screening.** Given ordering $\pi$ and scores $S$ from a backbone method, we construct ordering-constrained candidate sets for each node $X_i$:

**Definition 3.1** (Ordering-Constrained Candidate Sets). Let $\pi$ be a (possibly noisy) topological ordering and $S$ be backbone edge scores. For each node $X_i$, define:

$$\mathcal{C}_{\mathrm{Pa}}(X_i) = \mathrm{Top\text{-}K}(\{X_j : \pi(j) < \pi(i), \ S_{j,i} \geq \tau\}), \quad (1)$$
$$\mathcal{C}_{\mathrm{Ch}}(X_i) = \{X_k : X_i \in \mathcal{C}_{\mathrm{Pa}}(X_k)\}. \quad (2)$$

(Optional) Spouse closure augments the search space with candidate spouses:

$$\mathcal{C}_{\mathrm{Sp}}(X_i) = \left( \bigcup_{X_k \in \mathcal{C}_{\mathrm{Ch}}(X_i)} \mathcal{C}_{\mathrm{Pa}}(X_k) \right) \setminus \{X_i\}. \quad (3)$$

The candidate Markov blanket search space is:

$$\mathcal{C}_{\mathrm{MB}}(X_i) = \mathcal{C}_{\mathrm{Pa}}(X_i) \cup \mathcal{C}_{\mathrm{Ch}}(X_i) \cup \mathcal{C}_{\mathrm{Sp}}(X_i). \quad (4)$$

Here $\mathrm{Top\text{-}K}(\cdot)$ selects the top-$K$ elements by score, and $\tau$ is the score threshold.

The spouse closure in Eq. (3) adds candidate co-parents of candidate children, which is sufficient to cover spouse variables in Markov blankets (Definition A.4). This construction reduces parent search from $O(d)$ to $O(K)$ per node.

**Candidate set sizes.** By construction, $|\mathcal{C}_{\mathrm{Pa}}(X_i)| \leq K$ for all $i$, while $|\mathcal{C}_{\mathrm{Ch}}(X_i)|$ and $|\mathcal{C}_{\mathrm{Sp}}(X_i)|$ are data-dependent. Empirical statistics are in Appendix H.

### 3.2. Theoretical Guarantees

We establish consistency under explicit assumptions:

**Assumption 3.2** (Standard Causal Discovery Assumptions). (i) Causal Markov condition: every variable is conditionally independent of its non-descendants given its parents in $\mathcal{G}^*$. (ii) Faithfulness: $X \perp\!\!\!\perp Y \mid \mathbf{Z}$ in the data if and only if $X \perp_{\mathcal{G}^*} Y \mid \mathbf{Z}$ (d-separation). (iii) Causal sufficiency: no latent confounders. (iv) Consistent CI test: the CI oracle is correct in the population limit.

**Assumption 3.3** (High-Recall Ordering). Let $\pi$ be an ordering over variables, and let $\mathrm{Pred}_i = \{X_j : \pi(j) < \pi(i)\}$ denote the ordering-feasible predecessors of $X_i$. For each variable $X_i$, its true parent set $\mathrm{Pa}_{\mathcal{G}^*}(X_i)$ is contained in the top-$K$ predecessors with high probability:

$$\mathbb{P}\left(\mathrm{Pa}_{\mathcal{G}^*}(X_i) \subseteq \mathrm{Top\text{-}K}(\mathrm{Pred}_i)\right) \geq 1 - \delta,$$

where $K \ll d$ and $\delta$ may depend on ordering quality. Perfect ordering ($\forall (X_j \to X_i) \in \mathcal{G}^* : \pi(j) < \pi(i)$) is sufficient but not necessary.

*Remark* 3.4 (Ordering as a Soft Constraint). Assumption 3.3 does not require $\pi$ to be correct or consistent with the underlying DAG. The ordering serves as a soft directional filter restricting candidate parent sets, thereby bounding conditioning-set dimensionality. Even with substantial inversion errors, OCMB remains robust if the ordering preserves high recall of true parents (see Appendix O).

**Assumption 3.5** (Top-$K$ Parent Recall). For each node $X_i$, the backbone scores satisfy:

$$\mathrm{Pa}_{\mathcal{G}^*}(X_i) \subseteq \mathrm{Top\text{-}K}(\{X_j \in \mathrm{Pred}_i : S_{j,i} \geq \tau\}), \quad (5)$$
$$S_{j,i} \geq \tau \quad \forall X_j \in \mathrm{Pa}_{\mathcal{G}^*}(X_i).$$

If $\mathrm{Pa}_{\mathcal{G}^*}(X_i) = \emptyset$, the condition is vacuous and $\mathcal{C}_{\mathrm{Pa}}(X_i) = \emptyset$.

**Theorem 3.6** (Candidate Parent Recall). *Under Assumptions 3.3 and 3.5, $\mathrm{Pa}_{\mathcal{G}^*}(X_i) \subseteq \mathcal{C}_{\mathrm{Pa}}(X_i)$ holds for all $i$ with probability at least $1 - \delta$.*

*Proof Sketch.* By Assumption 3.3, true parents are contained in the top-$K$ predecessors with probability at least $1 - \delta$. Assumption 3.5 ensures all true parents have scores $S_{j,i} \geq \tau$ and are retained by Top-$K$ selection. Thus the candidate set contains all true parents with probability at least $1 - \delta$. $\square$

**Scope and Robustness.** Theorem 3.6 relies on Top-$K$ parent recall—a screening assumption requiring true parents are not pushed out by selection. Under stronger score separability ($\min_{X_j \in \mathrm{Pa}} S_{j,i} > \tau \geq \max_{X_k \notin \mathrm{Pa}} S_{k,i}$), exact recovery follows. When assumptions are violated, some parents may be excluded (false negatives); however, even with substantial ordering errors, restricting candidate set size improves finite-sample CI reliability by bounding conditioning dimensions (see Appendix O for random-ordering ablations).

**Lemma 3.7** (Candidate MB Superset)**.** *Under Assumptions 3.3 and 3.5, and with spouse closure enabled in Eq.* (3)*, $\mathrm{MB}_{\mathcal{G}^*}(X_i) \subseteq \mathcal{C}_{\mathrm{MB}}(X_i)$ for all $i$.*

*Proof Sketch.* By Theorem 3.6, $\mathcal{C}_{\mathrm{Pa}}(X_i)$ contains all true parents. For any true child $X_c \in \mathrm{Ch}_{\mathcal{G}^*}(X_i)$, we have $X_i \in \mathrm{Pa}_{\mathcal{G}^*}(X_c) = \mathcal{C}_{\mathrm{Pa}}(X_c)$, hence $X_c \in \mathcal{C}_{\mathrm{Ch}}(X_i)$. Finally, for any spouse $X_s$ of $X_i$, there exists a child $X_c$ with $X_s \in \mathrm{Pa}_{\mathcal{G}^*}(X_c) = \mathcal{C}_{\mathrm{Pa}}(X_c)$; spouse closure adds $\mathcal{C}_{\mathrm{Pa}}(X_c)$ into $\mathcal{C}_{\mathrm{MB}}(X_i)$. $\square$

**Theorem 3.8** (Consistent MB Recovery)**.** *Under Assumption 3.2, and given that Assumptions 3.3–3.5 hold so that Lemma 3.7 applies, constrained IAMB recovers the true Markov blanket: $\widehat{\mathrm{MB}}_i \xrightarrow{p} \mathrm{MB}_{\mathcal{G}^*}(X_i)$.*

*Proof Sketch.* By Lemma 3.7, constrained IAMB operates on a superset of the true Markov blanket. Standard IAMB consistency (Tsamardinos et al., 2003b) applies: the forward phase includes all variables with non-zero CMI; the backward phase removes spurious inclusions. The ordering constraint restricts the search domain without altering CI correctness. With Theorem 3.6, orienting edges using $\mathcal{C}_{\mathrm{Pa}}(X_i)$ recovers the true parents under correct ordering. $\square$

The novelty lies not in IAMB consistency itself, but in **constructing valid candidate supersets efficiently through ordering constraints**—infeasible in high-dimensional or hub-dominated graphs otherwise.

**Framework Assumptions and Stage-I Role.** OCMB is a *framework*: Stage-I assumptions are inherited from the chosen ordering backbone (e.g., additive noise models for CaPS, score-based identifiability for SciNO). The theoretical guarantees in Theorems 3.6–3.8 are therefore *conditional*—they require that Stage I produces candidate parent sets with sufficiently high recall, but do not themselves assume any specific data-generating mechanism beyond Assumption 3.2. In small-to-medium settings, Stage-I computation (ordering

estimation) may dominate total runtime; OCMB's efficiency advantage becomes most pronounced when the downstream CI-test burden would otherwise be prohibitive (e.g., hub-heavy graphs at $d \geq 100$).

**Role of Spouse Closure.** Lemma 3.7 provides a *conservative sufficient condition* ensuring $\mathcal{C}_{\mathrm{MB}}(X_i) \supseteq \mathrm{MB}_{\mathcal{G}^*}(X_i)$. This is intentionally stronger than what Algorithm 1 strictly requires: the practical pipeline targets *directed parent set recovery* (line 11 intersects $\widehat{\mathrm{MB}}_i$ with $\mathcal{C}_{\mathrm{Pa}}(X_i)$), not full Markov blanket estimation. Under a high-quality ordering, edge directions are already determined by the ordering constraint, reducing the empirical need for spouse-based collider detection. Under a poor ordering, the dominant error source is parent recall deficit, not conditioning-set incompleteness. Thus, spouse closure is a conservative option that preserves the theoretical guarantee but is not empirically essential in either regime (see Appendix H for ablations).

**Random Ordering and Finite-Sample Effects.** When Assumption 3.3 is violated (e.g., random ordering), the exact-recovery guarantee of Theorem 3.6 no longer holds. The controlled decomposition in Table 7 confirms this: OCMB-random-ord achieves F1=0.027, consistent with theorem failure. However, on certain benchmarks (Appendix O), random ordering still yields nontrivial SHD—we interpret this as a *finite-sample regularization effect* from bounded candidate restriction, not as evidence that the theorem applies under violated assumptions.

### 3.3. Algorithm

**Complexity.** Let $M_i \triangleq |\mathcal{C}_{\mathrm{MB}}(X_i)|$ denote the size of the candidate search space for node $i$. The constrained IAMB procedure performs $O(M_i^2)$ conditional mutual information (CMI) evaluations for node $i$, leading to a total cost of

$$O\left(\sum_{i=1}^{d} M_i^2 \cdot T_{\mathrm{CMI}}(n)\right), \qquad (6)$$

where $T_{\mathrm{CMI}}(n)$ is the cost of a single CMI estimation (e.g., $T_{\mathrm{CMI}}(n) = O(n^2)$ for brute-force kNN-based CMI (Kraskov et al., 2004); reducible to $O(n \log n)$ with tree-based nearest-neighbor structures). By construction, $|\mathcal{C}_{\mathrm{Pa}}(X_i)| \leq K$, while $M_i$ is data-dependent due to $\mathcal{C}_{\mathrm{Ch}}$ and spouse closure. Empirically, we observe $M_i \ll d$ on hub-heavy graphs, which yields substantial speedups over unconstrained CI-based methods.

**Remark.** If $M_i = O(K)$ for all $i$ (as observed empirically in our target regimes), then the above reduces to $O(d \cdot K^2 \cdot T_{\mathrm{CMI}}(n))$, achieving a $(d/K)^2$ speedup over classical methods. A key consequence is that the backward phase operates on bounded conditioning sets ($|\widehat{\mathrm{MB}}_i| \leq M_i \ll d$), placing CI tests in a statistically favorable regime compared

**Algorithm 1** OCMB: Ordering-Constrained Markov Blankets

**Require:** Data $\mathbf{X} \in \mathbb{R}^{n \times d}$; thresholds $\alpha, \tau$; max parents $K$

**Ensure:** Estimated DAG $\hat{\mathcal{G}}$

1: **Stage 1: Directional Screening** — Run ordering backbone $\rightarrow \pi, S$
2: **for** each node $i$ **do**
3:    $\mathcal{C}_{\mathrm{Pa}}(X_i) \leftarrow$ Top-K($\{j : \pi(j) < \pi(i), S_{j,i} \geq \tau\}$)
4:    $\mathcal{C}_{\mathrm{Ch}}(X_i) \leftarrow \{k : i \in \mathcal{C}_{\mathrm{Pa}}(X_k)\}$;   $\mathcal{C}_{\mathrm{Sp}}(X_i) \leftarrow \bigcup_{k \in \mathcal{C}_{\mathrm{Ch}}(X_i)} \mathcal{C}_{\mathrm{Pa}}(X_k) \setminus \{X_i\}$
5:    $\mathcal{C}_{\mathrm{MB}}(X_i) \leftarrow \mathcal{C}_{\mathrm{Pa}}(X_i) \cup \mathcal{C}_{\mathrm{Ch}}(X_i) \cup \mathcal{C}_{\mathrm{Sp}}(X_i)$ {Candidate search space}
6: **end for**
7: **Stage 2: Constrained IAMB** — For each node $i$:
8:    *(Grow)* **while** $\exists j \in \mathcal{C}_{\mathrm{MB}}(X_i) \setminus \widehat{\mathrm{MB}}_i$ s.t. $I(X_i; X_j \mid \widehat{\mathrm{MB}}_i) \geq \alpha$:
9:       $\widehat{\mathrm{MB}}_i \leftarrow \widehat{\mathrm{MB}}_i \cup \{\arg\max_j I(X_i; X_j \mid \widehat{\mathrm{MB}}_i)\}$
10:   *(Prune)* **for each** $j \in \widehat{\mathrm{MB}}_i$: remove $j$ if $I(X_i; X_j \mid \widehat{\mathrm{MB}}_i \setminus \{j\}) < \alpha$
11: **Edge Construction:** $A_{j,i} \leftarrow 1$ for $j \in \widehat{\mathrm{MB}}_i \cap \mathcal{C}_{\mathrm{Pa}}(X_i)$; **return** DAG from $A$

to unconstrained methods where hub nodes may induce conditioning sets of size $O(\Delta)$.

# 4. Experiments

Our experiments are designed to examine whether ordering-based directional constraints improve the statistical reliability of CI-based causal discovery, particularly in hub-dominated graphs. We focus on: (1) accuracy compared to baselines, (2) structure-dependent performance, (3) computational efficiency and scalability, and (4) robustness to key hyperparameters (Appendix H).

## 4.1. Datasets

(i) Synthetic data: We generate directed acyclic graphs using two standard random graph models: scale-free (SF, hub-dominated) networks following the Barabási–Albert preferential attachment model (Barabási & Albert, 1999), which exhibit heterogeneous degree distributions common in biological networks, and Erdős–Rényi (ER) random graphs (Erdős & Rényi, 1959) with uniform edge probability. Our main synthetic setup uses nonlinear structural equation models with Gaussian noise and $n$=1000 samples (Section 4.2); additional configurations are in Appendix H. (ii) Sachs: The Sachs protein signaling dataset (Sachs et al., 2005) contains $n$=1000 observational samples of $d$=11 phosphoproteins from human T-cells, with ground-truth established through literature review and perturbation ex-

periments. (iii) DREAM3/4: The DREAM3 (Prill et al., 2010) and DREAM4 (Marbach et al., 2010) challenges provide gene regulatory network benchmarks with $d$=50 and $d$=100 nodes respectively, derived from *E. coli* and *S. cerevisiae* transcriptional networks. (iv) Alarm: The Alarm network (Beinlich et al., 1989) is a standard Bayesian network benchmark with $d$=37 nodes representing medical diagnosis variables.

## 4.2. Experimental Setup

We generate scale-free (SF) (Barabási & Albert, 1999) and Erdős–Rényi (ER) (Erdős & Rényi, 1959) graphs with average degree 3 and $d$=100 nodes. Data are generated from nonlinear structural equation models with Gaussian noise; sample size is $n$=1000. Results are averaged over 10 random seeds. Baselines include constraint-based (PC (Spirtes et al., 2000), using the CUDA-accelerated GPUCSL implementation (Braun et al., 2022)), score-based (GRaSP (Lam et al., 2022)), and continuous optimization (DAGMA-linear (Bello et al., 2022)) methods. Unless noted, OCMB uses CaPS (Xu et al., 2024) as the ordering backbone. Metrics: SHD (lower=better), F1 (higher=better), runtime (seconds), and CI test count for CI-based methods. We use $K$=0.05$d$ (i.e., $K$=5), score threshold at 95th percentile ($\tau$=0.95), CI threshold $\alpha$=0.01, and enable spouse closure; sensitivity to $K$ is analyzed in Section 4.7, with extended results in Appendix H.

## 4.3. Main Results

*Table 1.* Performance on **Scale-free** graphs ($d$=100, $n$=1000, nonlinear + Gaussian, 10 seeds). Best in **bold**, second-best underlined. #CI Tests measures Stage-II local CI burden, not end-to-end cost.

| Method | SHD↓ | F1↑ | Time (s)↓ | #CI Tests↓ |
|---|---|---|---|---|
| DAGMA-linear | **45.0**±1.0 | **0.736**±0.01 | 20.0 | – |
| **OCMB-HITON** | 60.3±9.3 | 0.639±0.07 | 3.6 | **704** |
| H2PC | 64.3±11.5 | 0.550±0.11 | 108.0 | 32,178 |
| MMHC | 64.3±11.5 | 0.550±0.11 | 73.5 | 23,229 |
| OCMB | 65.3±3.5 | 0.542±0.03 | 3.6 | 572 |
| IAMB | 99.0±0.0 | 0.480±0.03 | 65.0 | 23,162 |
| GRaSP | 103.0±6.6 | 0.337±0.18 | 83.8 | – |
| PC | 158.7±23.4 | 0.427±0.05 | 1.9 | 3,413,957 |
| NOTEARS-MLP | 210.7±42.0 | 0.204±0.03 | 5.1 | – |
| CAM | 422.3±2.1 | 0.316±0.00 | 270.6 | – |
| CaPS-alone | 446.0±6.0 | 0.249±0.01 | 1.8 | – |

Table 1 presents main results on scale-free graphs, which exhibit hub-dominated structures common in biological networks. DAGMA-linear achieves the best accuracy (F1=0.736, SHD=45); notably, DAGMA-linear assumes a linear SEM that is not matched to our nonlinear data-generating process, so its strong performance is an empirical observation—plausibly reflecting a favorable inductive bias from its sparsity and acyclicity regularization on hub-dominated graphs. OCMB-HITON (OCMB with

HITON-MB (Aliferis et al., 2010) as the Stage-II learner) achieves the second-best accuracy (F1=0.639, SHD=60.3) with ~4850× fewer Stage-II CI tests than PC (704 vs. 3.4M). H2PC and MMHC achieve comparable accuracy (F1=0.550) but require substantially more CI tests (23k–32k). PC's relatively poor performance (SHD=158.7) confirms that high-dimensional conditioning around hub nodes leads to unreliable CI tests.

*Table 2.* Skeleton recovery on **Erdős–Rényi** graphs ($d$=100, $n$=1000, nonlinear + Gaussian, 10 seeds). Best in **bold**. OCMB-HITON achieves the best skeleton accuracy with ~11× fewer CI tests than PC.

| Method | $SHD_{skel}$ ↓ | $F1_{skel}$ ↑ | Time (s)↓ | #CI Tests↓ |
|---|---|---|---|---|
| **OCMB-HITON** | **77.3**±17.0 | **0.717**±0.04 | 7.6 | **2,072** |
| PC | 89.3±18.7 | 0.716±0.05 | 5.0 | 23,733 |
| GRaSP | 99.0±19.5 | 0.676±0.04 | 49.6 | – |
| DAGMA-linear | 103.3±18.3 | 0.624±0.04 | 22.1 | – |
| IAMB | 108.7±12.4 | 0.508±0.02 | 89.1 | 28,857 |
| OCMB | 112.0±13.7 | 0.488±0.04 | **6.8** | 1,576 |
| H2PC | 135.0±15.1 | 0.279±0.02 | 46.4 | 16,559 |
| MMHC | 135.0±15.1 | 0.279±0.02 | 49.9 | 19,089 |
| NOTEARS-MLP | 276.3±19.1 | 0.410±0.03 | 5.2 | – |
| CAM | 274.0±37.0 | 0.536±0.06 | 220.1 | – |
| CaPS-alone | 418.0±6.1 | 0.363±0.02 | 2.8 | – |

Table 2 shows skeleton recovery on ER graphs, where hub nodes are absent and conditioning-set explosion is less severe. We report skeleton metrics (undirected edges) rather than directed metrics, as direction identifiability varies across methods under nonlinear mechanisms. OCMB-HITON achieves the **best $F1_{skel}$** (0.717) and **lowest $SHD_{skel}$** (77.3), matching PC's skeleton accuracy ($F1_{skel}$=0.716) while requiring **11× fewer CI tests** (2,072 vs. 23,733). H2PC and MMHC perform poorly on ER graphs ($F1_{skel}$=0.279), suggesting sensitivity to graph structure. This delineates the regime of maximal benefit: OCMB is designed for hub-heavy scale-free graphs where it substantially improves both accuracy and efficiency; on homogeneous ER graphs, OCMB remains competitive with dramatically lower CI-test budget. However, OCMB's *directed* edge recovery on ER graphs is weaker (directed F1=0.236–0.355 vs. PC's 0.663; Appendix D), reflecting that ordering priors are less informative when hub structure is absent. We regard this as an explicit applicability boundary of the current method.

### 4.4. Stage II Replaceability

To verify that OCMB's gains stem from the framework design rather than a specific MB learner, we evaluate Stage II with alternative local structure learners: HITON-MB (Aliferis et al., 2010) and MMPC (Tsamardinos et al., 2003a), replacing the default IAMB.

Table 3 shows that all three OCMB variants substantially outperform baselines, with OCMB-HITON achieving the

*Table 3.* Stage II replaceability on SF graphs ($d$=100−200, $n$=1000). Different MB learners yield consistent improvements; HITON-MB performs best. OCMB variants achieve ~3000× fewer CI tests than PC.

| Method | F1↑ | SHD↓ | #CI Tests↓ |
|---|---|---|---|
| *Baselines:* | | | |
| PC | 0.483±0.03 | 212.7±83 | 4,464,596 |
| NOTEARS-MLP | 0.240±0.04 | 222.2±39 | – |
| *OCMB with different MB learners:* | | | |
| IAMB (default) | 0.489±0.06 | 124.7±61 | 1,158 |
| MMPC | 0.487±0.06 | 129.8±65 | 2,123 |
| **HITON-MB** | **0.542**±0.09 | **123.3**±66 | **1,038** |

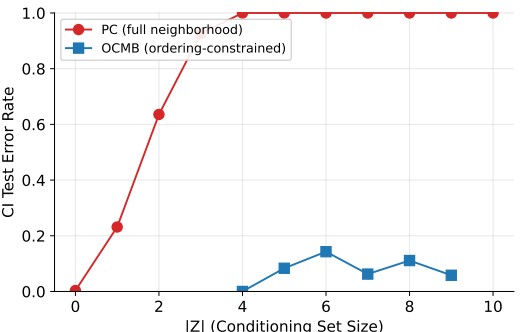

*Figure 3.* CI test error rate vs. conditioning set size on **scale-free** synthetic graphs. PC's error reaches 100% for $|Z| \geq 4$; OCMB stays below 15%, stabilizing at 6% for $|Z| = 9$ (based on 2,800 CI queries).

best accuracy (F1=0.542, SHD=123.3) using ~3000× fewer CI tests than PC. This confirms that OCMB's gains arise from the **ordering-constrained framework**, not from a particular MB algorithm—the Stage II learner is modular and replaceable.

### 4.5. CI Test Error Analysis

To validate that OCMB improves accuracy via reduced conditioning set dimensionality, we analyze CI error rates vs. $|Z|$ on scale-free graphs ($d = 100, 200$).

PC frequently operates in high-dimensional regimes due to hub nodes, leading to unreliable CI tests. OCMB maintains small conditioning sets by restricting to ordering-feasible predecessors, validating the core mechanism underlying its accuracy gains.

**Per-order error distribution.** To quantify the severity, we decompose PC's 3.4M CI tests (Table 1) by conditioning-set order on scale-free graphs ($d$=100): 97.9% of tests fall in the $|Z| \geq 4$ regime where the error rate is 100%, while only 0.2% are at order 0 (error =0%). OCMB avoids this entirely by construction: candidate sets are small (median $|\mathcal{C}_{MB}| \leq 5$), so all CI tests remain in the statistically reliable

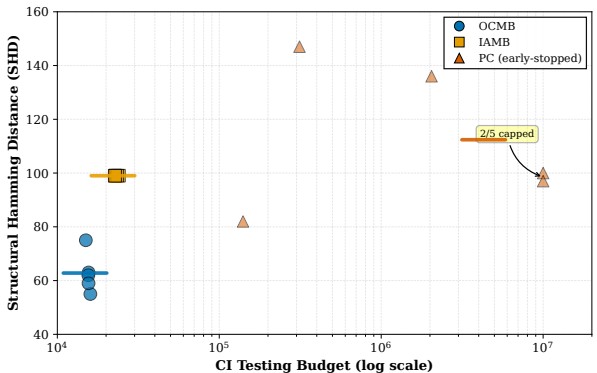

*Figure 4.* SHD vs. statistical query budget (log scale) on **scale-free** synthetic graphs ($d$=100, $n$=2000, nonlinear + Gaussian, 5 seeds). OCMB clusters tightly around 15k queries (CV=2.1%), while PC spans 140k–10M+ queries (CV=100.9%) with 2/5 runs hitting the 10M limit.

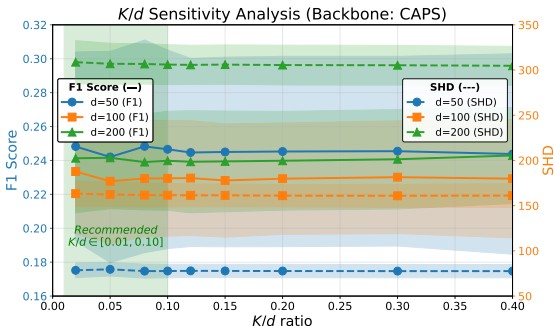

*Figure 5.* $K/d$ sensitivity analysis for OCMB(caps) on **Erdős–Rényi** synthetic graphs (nonlinear + Gaussian). F1 remains stable across a wide $K/d$ range: for $d$=50 and $d$=200, F1 varies by less than 4% from $K/d$=0.02 to 0.40. The green band marks the recommended range $K/d \in [0.05, 0.20]$.

low-order regime.

### 4.6. Statistical Query Budget Analysis

To isolate algorithmic efficiency from implementation details, we count statistical queries (each CI test $I(X; Y|S)$ counts as one unit) on scale-free graphs ($d = 100$, $n = 2000$, 5 seeds).

Figure 4 shows that OCMB achieves mean SHD of 62.8 with 15,545 queries, while PC requires 140k–10M+ queries (mean SHD 112.4). OCMB achieves 44% lower error with 289× fewer queries by restricting candidates per node from $O(d)$ to $O(K)$. Ordering constraints also provide predictability (CV=2.1% vs. PC's 100.9%), relevant for production deployment where resource budgets must be planned.

### 4.7. $K$ Sensitivity Analysis

A natural concern is whether the heuristic $K$=0.05$d$ is robust. We sweep $K/d \in \{0.02, 0.05, 0.10, 0.20, 0.40\}$ on ER graphs with $d \in \{50, 100, 200\}$.

Figure 5 reveals that **OCMB exhibits remarkable insensitivity to $K$**: for $d$=50 and $d$=200, F1 varies by less than 4% across $K/d \in [0.02, 0.40]$. We recommend $K^* = \max(2, \lceil 0.05d \rceil)$; extended analysis is in Appendix H.1.

### 4.8. Real-World Benchmark

We evaluate OCMB on established benchmarks: Sachs, DREAM3/4, and Alarm (Section 4.1), comparing against baselines from Dong & Gao (2025). Tables 1–3 report baselines we evaluated directly on our synthetic setup; Tables 4–6 follow the benchmark protocol of Dong & Gao (2025), with DAGMA-linear results added for completeness.

**Sachs.** OCMB-scino achieves the lowest SHD (21), outperforming EEMBI-PC (27) by 22%.

*Table 4.* Results on **Sachs** (SHD on DAG).

| Method | SHD↓ |
|---|---|
| *Baselines from Dong & Gao (2025):* | |
| DAGMA-linear (Bello et al., 2022) | 25 |
| EEMBI-PC | 27 |
| NOTEARS (Zheng et al., 2018) | 28 |
| EEMBI | 30 |
| PC | 32 |
| **OCMB-scino** | **21** |
| OCMB-caps | 23 |

**DREAM3.** OCMB variants substantially outperform all baselines, achieving 43–56% lower SHD than EEMBI.

*Table 5.* SHD on DREAM3 networks. OCMB reduces error by ~50% vs. EEMBI. Baselines from Dong & Gao (2025), with DAGMA-linear added for completeness.

| Method | Ecoli1 | Ecoli2 | Yeast1 | Yeast2 | Yeast3 |
|---|---|---|---|---|---|
| PC | 212 | 216 | 216 | 257 | 293 |
| NOTEARS | 157 | 181 | 195 | 230 | 253 |
| DAGMA-linear | 111 | 118 | 106 | 191 | 208 |
| DiffAN | 204 | 176 | 205 | 245 | 244 |
| EEMBI | 124 | 164 | 143 | 215 | 226 |
| EEMBI-PC | 144 | 158 | 158 | 220 | 236 |
| **OCMB-SciNO** | **70** | **93** | **86** | **169** | 194 |
| **OCMB-CaPS** | 71 | 95 | 91 | 177 | **190** |

**DREAM4 and Alarm.** On DREAM4, OCMB-caps achieves mean SHD of 221, outperforming PC (327) by 32%, NOTEARS (794) by 72%, and DAGMA-linear (mean SHD 278) by 20%. On Alarm, OCMB-scino achieves SHD of 44, compared to PC (53), NOTEARS (47), and DAGMA-linear (57).

*Table 6.* Results on **DREAM4** (100 nodes) and **Alarm** (37 nodes). See Appendix O for Oracle and Random orderings.

| Method | Net1 | Net2 | Net3 | Net4 | Net5 | Alarm |
|---|---|---|---|---|---|---|
| PC | 305 | 385 | 305 | 332 | 307 | 53 |
| NOTEARS | 704 | 913 | 691 | 868 | 793 | 47 |
| DAGMA-linear | 243 | 303 | 267 | 296 | 282 | 57 |
| **OCMB-caps** | **187** | **263** | **214** | **229** | **213** | 49 |
| **OCMB-scino** | 200 | 286 | 210 | 253 | 227 | **44** |

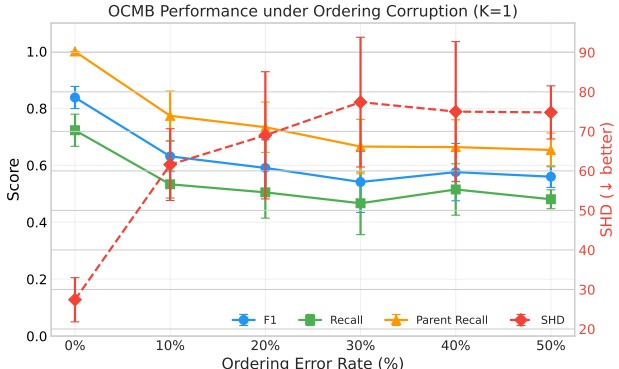

*Figure 6.* OCMB under ordering corruption on **scale-free** synthetic graphs ($d=100$, $n=1000$, nonlinear + Gaussian, $K=1$). With 50% nodes shuffled, OCMB retains F1≈0.56, demonstrating graceful degradation.

**Remark.** Sachs ($d=11$) provides limited benefit from ordering constraints due to low dimensionality. OCMB's advantage is most pronounced on medium/large graphs (DREAM3/4, Alarm) where hub nodes induce high-dimensional conditioning in constraint-based methods.

### 4.9. Ordering Robustness Analysis

A key concern for ordering-based methods is sensitivity to ordering quality. We systematically evaluate OCMB's robustness by injecting controlled errors into the oracle ordering: starting from the true topological ordering, we randomly shuffle $x\%$ of nodes' relative positions.

Figure 6 shows the results. With perfect ordering (0% error), OCMB achieves F1=0.84 and 100% Precision. As error rate increases to 50%, F1 degrades gracefully to 0.56 (33% drop), and Parent Recall drops from 1.0 to 0.65. The key insight is that **OCMB exhibits graceful degradation**: even with substantially corrupted orderings, performance remains practical rather than collapsing. This addresses concerns about ordering dependency—OCMB does not require perfect orderings; moderate corruption still yields stable results, and degradation is predictable rather than catastrophic. The controlled decomposition in Table 7 further shows that moderate backbone recall ($\text{Rec}_{\text{Pa}}=0.57$ for CaPS) suffices for substantial gains over both PC and the backbone alone. Moreover, an ordering-variant ablation on DREAM4 and Alarm (Appendix O) confirms consistent benefit across CaPS, SciNO, and even random orderings. Extended analysis across different $K$ values is provided in Appendix N.

### 4.10. Scalability

OCMB scales linearly up to $d=200$ (while PC times out in 40% of runs). A preliminary experiment at $d=500$ (1/5 seeds successful on SF graphs, 0/5 on ER) suggests feasibility of the approach at larger scales, though the evidence is statistically limited; detailed results are in Appendix K.

## 5. Discussion

### 5.1. Practical Implications for High-dimensional Settings

The preliminary $d=500$ experiment (1/5 seeds successful on SF graphs; Appendix M) offers initial evidence that the OCMB design can scale beyond $d=200$. In the single successful seed, the ordering backbone alone fails catastrophically (SHD ≈12,361), while OCMB reduces SHD to 328. However, the low success rate precludes strong conclusions about scalability; we present this as preliminary evidence of feasibility rather than a robust scaling result. Further engineering improvements—approximate nearest-neighbor search, CI test caching, and GPU-accelerated estimators—would be needed for routine application at this scale.

The key insight is that ordering constraints encode global directional information, restricting parent candidates to predecessors. Around hub nodes with degree $\Delta$ in scale-free graphs, skeleton-based methods require conditioning sets of size $O(\Delta)$, while OCMB conditions only on other candidate parents, bounded by in-degree $k \ll \Delta$. This explains OCMB's ~2.6× SHD reduction over PC on scale-free graphs (SHD 60.3 vs. 158.7) and its reduced advantage on homogeneous ER graphs, where hub-induced dimensionality problems are less severe.

### 5.2. Component Synergy

Neither ordering backbone alone (high false positives, SHD≈112) nor IAMB alone (lower F1, 3× more CI tests) matches OCMB's performance. The ordering provides directional screening and candidate construction (Theorem 3.6 and Lemma 3.7), while CI testing provides local statistical validation (Theorem 3.8). This demonstrates the necessity of both components.

*Table 7.* Decomposing Stage-I: directionality vs. sparsification. $\text{Rec}_{\text{Pa}}$: candidate parent recall; med |CandMB|: median candidate set size.

| Variant | $\text{Rec}_{\text{Pa}} \uparrow$ | med |CandMB| $\downarrow$ | F1 $\uparrow$ |
|---|---|---|---|
| **OCMB-oracle** | **1.00** | **3.0** | **0.834** |
| OCMB-caps (ours) | 0.57 | 5.1 | 0.437 |
| OCMB-random-ord | 0.14 | 55.5 | 0.027 |
| OCMB-random-score | 0.16 | 47.4 | 0.039 |
| OCMB-pure-random | 0.07 | 48.2 | 0.016 |

### 5.3. Sparsity vs Directionality: Disentangling Sources of Improvement

An important question is whether OCMB's gains stem primarily from directional correctness (ordering parents before children) or from sparsity constraints (bounding candidate set sizes). We design a controlled decomposition with five variants that systematically vary the ordering $\pi$ and score matrix $S$ in Stage I (Table 7): (1) OCMB-oracle uses ground-truth topological ordering with backbone scores; (2) OCMB-caps uses learned CaPS ordering; (3) OCMB-random-ord uses random permutation with backbone scores; (4) OCMB-random-score uses ground-truth ordering with random $U(0, 1)$ scores; and (5) OCMB-pure-random uses random ordering with random scores.

Both ordering quality and score informativeness prove essential. Without correct ordering (OCMB-random-ord), F1 collapses to 0.027 even with informative scores; without informative scores (OCMB-random-score), candidate sets balloon (|CandMB| $\approx$ 47) and F1 drops to 0.039. This decomposes Stage I into two orthogonal effects: (i) dimensionality control via bounded candidate sets, and (ii) directional informativeness via parent recall. Both are necessary.

### 5.4. Limitations and Extensions

(i) Ordering quality dependency: OCMB's guarantees rely on the backbone producing a correct ordering (and informative scores). **If true parents are excluded from the Top-$K$ candidate set, recovery of those parent-child relationships is impossible**—this is OCMB's fundamental failure mode. When edges are misordered, OCMB may exclude true parents from candidates. However, as demonstrated in Section 4.9, this leads to graceful degradation rather than catastrophic failure: with 50% ordering corruption, Parent Recall drops to 0.65 and F1 degrades by 33%. (ii) Causal sufficiency: OCMB assumes causal sufficiency (no latent confounders). The ordering-constrained formulation is orthogonal to latent-variable handling and could incorporate FCI-style rules in future work. (iii) Parameter sensitivity: OCMB has hyperparameters $\tau$ (score threshold), $\alpha$ (CI threshold), and $K$ (max parents). Because $\alpha$ is applied to an estimated CMI statistic rather than a cali-

brated $p$-value, its effective scale depends on sample size $n$ and conditioning dimension $|S|$. Practitioners should perform sensitivity sweeps (e.g., $\alpha \in \{0.005, 0.01, 0.02\}$) and, when computation allows, use permutation-based calibration (Appendix F). The fixed $K$ represents an inductive bias controlling bias–variance trade-off: larger $K$ increases parent recall but expands conditioning sets, while smaller $K$ reduces variance at the cost of potential false negatives. An adaptive $K$ would require reliable estimation of node-specific in-degree, which itself relies on accurate CI testing—creating a circular dependency. Fixing $K$ breaks this circularity and provides a stable inductive bias. Sensitivity analysis is reported in Appendix H. (iv) Computational cost at very large scales: While our $d=500$ experiments provide preliminary evidence of feasibility, the computational cost (3245s, 373k CI tests on SF graphs) and low success rate (1/5 seeds) indicate that further engineering optimizations would be beneficial for routine application at industrial scales. The dominant bottleneck is the exact kNN-based CMI computation, which requires all-pairs distance scans and repeated neighbor counting in marginal subspaces. (v) Backbone assumptions: Stage-I ordering backbones (CaPS, SciNO) assume additive noise models (ANMs). Our theory does not guarantee robustness to ANM violations; the competitive benchmark performance on Sachs, DREAM3/4, and Alarm is empirical evidence of practical utility under approximate model match, not a controlled stress test under explicit ANM violations.

**Assumption Flexibility.** OCMB introduces a computational–statistical trade-off with a modular assumption structure. Stage I introduces additional assumptions localized to the ordering backbone (e.g., ANM for CaPS); Stage II retains the full flexibility of constraint-based methods—any CI test (e.g., kernel-based, copula-based) can be used. This modularity ensures that OCMB benefits from future ordering methods with weaker assumptions (e.g., interventional or domain-informed approaches) without modifying the local refinement stage.

## 6. Conclusion

We proposed OCMB, a two-stage causal discovery framework that decouples global directional estimation from local statistical validation. By leveraging causal orderings as soft constraints to restrict candidate parent sets, OCMB ensures all CI tests operate under bounded conditioning sets, addressing a fundamental limitation of constraint-based methods in hub-dominated graphs. Empirically, OCMB achieves substantial gains in structural accuracy (43–56% lower SHD than state-of-the-art on DREAM3) while using orders of magnitude fewer CI queries. Future work includes extending OCMB to settings with latent confounders and developing adaptive mechanisms for ordering quality estimation.

## Acknowledgements

This work was supported in part by the National Key R&D Program of China under Grant No. 2023YFC2508704, in part by the National Natural Science Foundation of China under Grant No. 62236008, in part by the Natural Science Foundation of Beijing under Grant No. L251082, and in part by the Shandong Provincial Natural Science Foundation under project ZR2025ZD01.

## Impact Statement

This paper presents work whose goal is to advance the field of Machine Learning, specifically causal discovery methods. Causal discovery has broad applications in science, medicine, and policy-making. There are many potential societal consequences of our work, none of which we feel must be specifically highlighted here.

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

## A. Standard Assumptions

**Definition A.1** (Faithfulness). A distribution $P$ is *faithful* to DAG $\mathcal{G}$ if every conditional independence in $P$ corresponds to d-separation in $\mathcal{G}$: $X \perp\!\!\!\perp_P Y \mid \mathbf{Z} \Leftrightarrow X \perp_\mathcal{G} Y \mid \mathbf{Z}$.

**Definition A.2** (Causal Sufficiency). A set of variables $\mathcal{V}$ is *causally sufficient* if for any $X_i, X_j \in \mathcal{V}$, all common causes are in $\mathcal{V}$ (no latent confounders).

**Definition A.3** (Additive Noise Model). $X_i = f_i(\text{Pa}_{\mathcal{G}^*}(X_i)) + E_i$ where noise $E_i$ enters additively and is independent of parents.

**Definition A.4** (Markov Blanket). $\text{MB}_\mathcal{G}(X_i) = \text{Pa}_\mathcal{G}(X_i) \cup \text{Ch}_\mathcal{G}(X_i) \cup \text{CoParents}_\mathcal{G}(X_i)$ is the minimal set rendering $X_i$ conditionally independent of all other variables.

**Definition A.5** (Topological Ordering). A permutation $\pi$ is a topological ordering of $\mathcal{G}$ if $(X_j \to X_i) \in \mathcal{E} \Rightarrow \pi(j) < \pi(i)$.

## B. Runtime and CI Test Efficiency Comparison

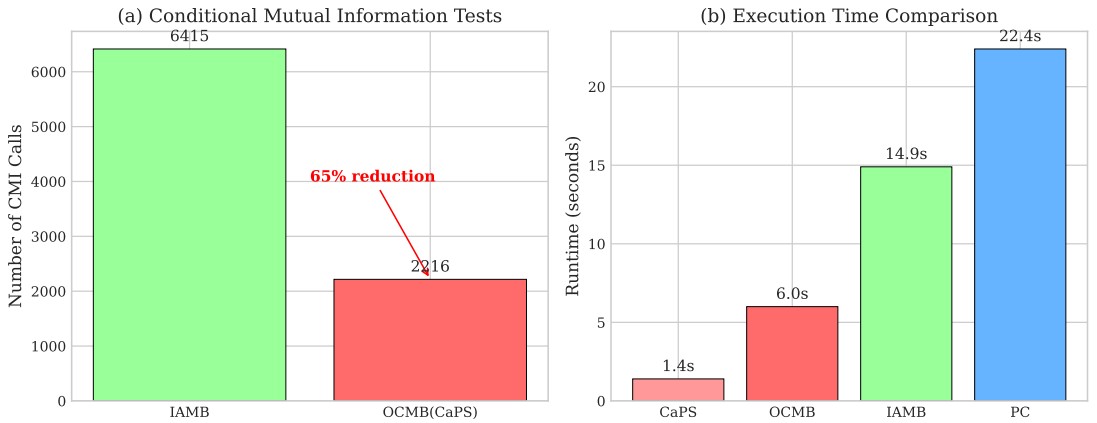

*Figure 7.* Runtime and CI test comparison across methods. OCMB achieves $2\times$ faster runtime and 65% fewer CI tests than IAMB while maintaining competitive accuracy.

## C. CMI Estimation Details

We use the kNN-based estimator of Kraskov et al. (2004):

$$\hat{I}(X; Y \mid \mathbf{Z}) = \psi(k) - \frac{1}{n}\sum_{i=1}^{n}\left[\psi(n_{xz}^{(i)}) + \psi(n_{yz}^{(i)}) - \psi(n_z^{(i)})\right] \tag{7}$$

where $\psi$ is the digamma function, $k$ is the number of nearest neighbors, and $n_{xz}^{(i)}, n_{yz}^{(i)}, n_z^{(i)}$ are local neighbor counts.

## D. Directed Edge Recovery on Erdős–Rényi Graphs

Table 8 shows directed edge recovery metrics on ER graphs, complementing the skeleton results in the main text (Table 2). PC achieves the best directed F1 (0.663), followed by GRaSP (F1=0.656). IAMB achieves F1=0.400, while H2PC and MMHC perform poorly on ER graphs (F1=0.250), suggesting these hybrid methods are sensitive to graph structure. OCMB's lower directed F1 (0.236–0.355) reflects that ordering-based direction constraints may introduce false negatives when the backbone ordering is imperfect; however, as shown in Table 2, skeleton recovery remains competitive with dramatically fewer CI tests.

## E. H2PC and MMHC Implementation Notes

H2PC and MMHC show mixed performance across graph types. On scale-free graphs, they achieve competitive accuracy (F1=0.550, SHD=64.3), ranking behind DAGMA-linear and OCMB-HITON but ahead of PC. However, on Erdős–Rényi graphs, their performance degrades substantially ($F1_{\text{skel}}=0.279$), suggesting sensitivity to graph structure.

*Table 8.* Directed edge recovery on **Erdős–Rényi** graphs ($d=100$, $n=1000$, nonlinear + Gaussian, 10 seeds). Best in **bold**. PC achieves the best directed F1 on ER graphs.

| Method | SHD↓ | F1↑ | #CI Tests↓ |
|---|---|---|---|
| GRaSP | **105.0**±19.5 | **0.656**±0.04 | – |
| PC | 109.0±21.0 | 0.663±0.05 | 23,733 |
| H2PC | 140.3±15.5 | 0.250±0.01 | 16,559 |
| MMHC | 140.3±15.5 | 0.250±0.01 | 19,089 |
| OCMB | 167.3±17.6 | 0.236±0.03 | **1,576** |
| IAMB | 168.3±11.8 | 0.400±0.01 | 28,857 |
| OCMB-HITON | 175.3±19.3 | 0.355±0.02 | 2,072 |
| DAGMA-linear | 192.0±27.2 | 0.300±0.04 | – |
| CAM | 284.0±35.1 | 0.519±0.05 | – |
| NOTEARS-MLP | 371.0±18.5 | 0.275±0.03 | – |
| CaPS-alone | 540.0±16.5 | 0.177±0.01 | – |

This structure-dependent behavior may stem from the greedy forward-backward skeleton discovery phase (MMPC), which can be affected by the degree distribution. On ER graphs with homogeneous degrees, the greedy adding strategy may prematurely terminate, missing true edges. OCMB avoids this issue by using ordering constraints to restrict the candidate set before CI testing.

Despite requiring 16k–32k CI tests (compared to OCMB's 572–2,072), H2PC and MMHC do not consistently outperform OCMB across graph types. OCMB offers more robust performance with substantially lower computational cost.

## F. Limitations

1. **Ordering Quality Dependency**: OCMB's guarantees rely on the backbone producing a correct ordering (and informative scores). **If true parents are excluded from the Top-$K$ candidate set, recovery of those parent-child relationships is impossible**—this is OCMB's fundamental failure mode. We quantify this effect in Section 4.9: with 50% ordering corruption, Parent Recall drops to 0.65 and F1 degrades by 33%, demonstrating graceful rather than catastrophic degradation.

2. **Parameter Sensitivity**: OCMB has hyperparameters $\tau$ (score threshold), $\alpha$ (CI threshold), and $K$ (max parents). Because $\alpha$ is applied to an estimated CMI statistic (not a calibrated $p$-value), its scale can depend on $n$ and the conditioning dimension; adaptive selection and statistical calibration remain future work. For practitioners applying OCMB to new datasets, we recommend: (1) start from the default $\alpha=0.01$; (2) run a small sensitivity sweep (e.g., $\alpha \in \{0.005, 0.01, 0.02\}$) and check edge stability; (3) if computation allows, use permutation-based calibration to estimate a null scale for the target sample size and conditioning dimensions.

3. **Assumption Violations**: Like most methods, OCMB assumes causal sufficiency and faithfulness. Extensions for latent confounders (e.g., FCI-style rules) remain future work.

4. **Non-identifiability**: Under non-additive noise or faithfulness violations, even perfect ordering and CI testing may not recover the true graph.

## G. Applicable Scenarios

OCMB is recommended for:

- **Scale-free networks**: Biological, social, citation networks with heterogeneous degree distributions

- **Medium-scale problems**: 20–200 nodes (with preliminary evidence at $d=500$; see Appendix K)

- **Sparse structures**: Average in-degree $< 10$

- **Continuous data**: Current implementation uses kNN-based CI tests

- **High-precision requirements**: Applications where false positives are costly

For homogeneous networks, PC may remain competitive. For very large-scale problems ($d > 1000$), pure ordering methods may be necessary despite lower precision.

## H. Hyperparameter Sensitivity

### H.1. $K$ Sensitivity and Adaptive Selection

A natural concern is that the heuristic $K \approx 0.05d$ is coarse and lacks an adaptive mechanism. We conduct a systematic sensitivity analysis by sweeping $K/d \in \{0.02, 0.05, 0.08, 0.10, 0.12, 0.15, 0.20, 0.30, 0.40\}$ on ER graphs with $d \in \{50, 100, 200\}$ using CAPS backbone (10 seeds for $d$=50, 3–4 seeds for $d \geq 100$).

**Key Finding: Robustness Plateau.** Table 9 reveals a striking pattern: **OCMB exhibits remarkable insensitivity to $K$ across a wide range**. For $d$=50 and $d$=200, F1 varies by less than 4% across all tested $K/d$ values from 0.02 to 0.40. The $d$=100 case shows somewhat larger sensitivity, but even there the degradation is gradual rather than catastrophic.

*Table 9.* $K/d$ sensitivity analysis for OCMB(caps) on ER graphs. $\star$ marks optimal; $\diamond$ marks within 3% of optimal. OCMB shows remarkable robustness across $K/d$ values.

| $d$ | $K/d$ | $K$ | F1↑ | $\Delta$F1 | SHD↓ | #CI Tests |
|---|---|---|---|---|---|---|
|  | 0.02 | 1 | 0.248$\diamond$ | 0.0% | 78.5 | 327 |
|  | 0.08 | 4 | 0.248$\star$ | 0.0% | 77.6 | 2,429 |
| 50 | 0.10 | 5 | 0.247$\diamond$ | 0.4% | 77.7 | 3,225 |
|  | 0.20 | 10 | 0.245$\diamond$ | 1.2% | 77.7 | 6,202 |
|  | 0.40 | 20 | 0.244$\diamond$ | 1.6% | 77.6 | 8,505 |
|  | 0.02 | 2 | 0.222$\star$ | 0.0% | 165.8 | 1,854 |
|  | 0.05 | 5 | 0.216$\diamond$ | 2.7% | 164.5 | 7,541 |
| 100 | 0.08 | 8 | 0.214 | 3.6% | 164.5 | 13,903 |
|  | 0.10 | 10 | 0.203 | 8.6% | 169.0 | 17,532 |
|  | 0.20 | 20 | 0.204 | 8.1% | 167.7 | 28,814 |
|  | 0.02 | 4 | 0.235$\diamond$ | 2.9% | 306.2 | 11,295 |
|  | 0.05 | 10 | 0.235$\diamond$ | 2.9% | 304.5 | 42,890 |
| 200 | 0.10 | 20 | 0.233$\diamond$ | 3.7% | 303.2 | 85,007 |
|  | 0.20 | 40 | 0.234$\diamond$ | 3.3% | 303.2 | 125,034 |
|  | 0.40 | 80 | 0.242$\star$ | 0.0% | 300.0 | 147,550 |

**Interpretation: Why $K$ Choice is Not Critical.** The robustness plateau admits a principled explanation:

1. **Backbone quality dominates**: When the ordering backbone (CAPS) is accurate, most true parents rank highly in the candidate set. Even small $K$ captures the essential parent-child relationships, while larger $K$ merely adds marginal candidates that are correctly filtered by CI tests.

2. **CI testing provides safety net**: Regardless of $K$, the subsequent CI testing phase removes false candidates. The main cost of large $K$ is computational (more CI tests), not accuracy degradation.

3. **Bounded conditioning set**: Unlike PC which may condition on arbitrarily large sets, OCMB's conditioning set is bounded by the candidate set size, maintaining CI test reliability.

**Practical Guideline.** Based on these findings, we recommend the simple heuristic:

$$K^* = \max\left(2, \lceil 0.05d \rceil\right) \tag{8}$$

This provides a reasonable balance between computational cost and accuracy. Importantly, **the exact choice of $K$ is not critical**: any value in the range $K/d \in [0.05, 0.20]$ yields near-optimal performance across all tested dimensions. This robustness is a practical advantage: users need not fine-tune $K$ for their specific dataset.

**Visualization.** Figure 8 provides a multi-panel visualization of the sensitivity analysis, showing F1 and SHD separately for each dimension. The key takeaway is that F1 remains stable across a wide $K/d$ range, with the main variation coming from random seed differences rather than $K$ choice.

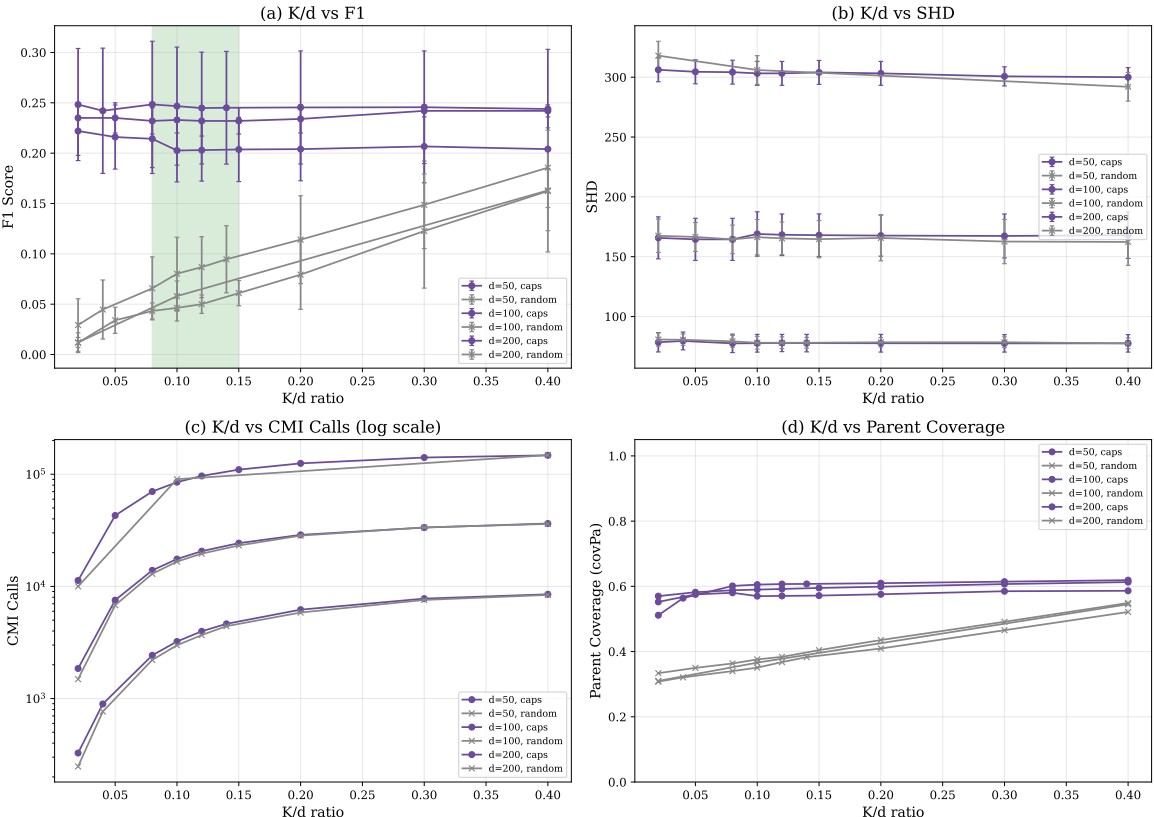

*Figure 8.* $K/d$ sensitivity analysis for OCMB(caps), multi-panel view. Each row shows a different dimension ($d$=50, 100, 200). Left column: F1 score; right column: SHD. All dimensions show robustness plateaus where F1 varies by less than 5% across the tested range.

**Comparison with Random Backbone.** Table 10 compares CAPS vs. random backbone across $K/d$ values. CAPS consistently outperforms random by 2–4× in F1, confirming that **backbone quality matters more than $K$ tuning**. Even with aggressive $K$ restriction, random backbone cannot match CAPS performance.

*Table 10.* Backbone quality vs. $K$ selection ($d$=50). CAPS dominates random across all $K/d$.

| $K/d$ | F1 (caps) | F1 (random) | Ratio | #CI Tests |
|------|-----------|-------------|-------|-----------|
| 0.02 | 0.248 | 0.029 | 8.5× | 327 |
| 0.10 | 0.247 | 0.080 | 3.1× | 3,225 |
| 0.40 | 0.244 | 0.186 | 1.3× | 8,505 |

**$\tau$ Sensitivity.** We vary the score threshold quantile $\tau$ (with $K/d = 0.2$) to study the accuracy–cost trade-off. Table 11 shows that $\tau$=0.85 balances accuracy and efficiency.

**Spouse Closure.** Enabling spouse closure increases the candidate search cost with minor changes in accuracy. Table 12 shows the trade-off: spouse closure doubles CI tests with negligible accuracy gain.

**Redundancy of Spouse Selection under Directional Priors.** This result may appear counterintuitive, as spouses are part of the classical Markov blanket definition. The explanation lies in OCMB's orientation mechanism: OCMB *does not primarily rely on V-structure discovery for edge orientation*. Instead, edge directions are determined by the ordering constraint ($\pi(j) < \pi(i)$ for $X_j \rightarrow X_i$). The ordering prior effectively resolves the Markov equivalence class to a single DAG (or a smaller set), reducing the empirical need for spouse-based collider detection. In classical constraint-based methods, spouses are essential for detecting collider structures ($X_i \rightarrow X_k \leftarrow X_j$), which provide orientation information through V-structures. However, OCMB obtains orientation directly from the ordering prior. In practice, parents and children

*Table 11.* $\tau$ ablation for OCMB(caps) on ER ($d = 100$).

| $\tau$ (quantile) | F1$_{skel}$ ↑ | SHD$_{skel}$ ↓ | Time (s) | #CI Tests |
|---|---|---|---|---|
| 0.70 | 0.435 | 115.4 | 21.5 | 5567 |
| 0.85 | 0.451 | 113.4 | 8.1 | 1605 |
| none | 0.428 | 116.6 | 70.7 | 20359 |

often suffice for accurate edge recovery under ordering constraints, though spouse closure remains a useful conservative option to preserve the MB superset guarantee of Lemma 3.7. We do not claim that spouses are universally redundant; rather, the theoretical condition is intentionally stronger than what the practical pipeline requires.

*Table 12.* Spouse closure ablation for OCMB(caps).

| Setting | F1$_{skel}$ ↑ | SHD$_{skel}$ ↓ | Time (s) | #CI Tests |
|---|---|---|---|---|
| spouse=off | 0.445 | 114.5 | 27.3 | 7400 |
| spouse=on | 0.431 | 116.1 | 56.9 | 16259 |

## I. Component Ablation Study

To understand the contribution of each component in OCMB, we compare against ordering-only (CaPS-alone) and local refinement-only (IAMB) baselines on scale-free graphs with Gaussian noise and mixed functional forms ($n = 1000$).

*Table 13.* Component ablation on SF + Gauss + mixed ($n = 1000$). Both ordering and local refinement are necessary for optimal performance.

| Configuration | SHD↓ | F1↑ | #CI Tests |
|---|---|---|---|
| CaPS-alone (ordering only) | 112.0 | 0.349 | – |
| IAMB (local refinement only) | 49.0 | 0.483 | 5699 |
| **OCMB (ordering + local)** | **31.7** | **0.558** | **1980** |

**Analysis.** Both components contribute essential benefits:

- **Ordering constraints** improve F1 from 0.483 (IAMB) to 0.558 (OCMB), a 16% relative improvement, by restricting candidate parents to ordering-feasible predecessors.

- **Local CI refinement** reduces SHD from 112 (CaPS-alone) to 31.7 (OCMB), a 72% reduction, by filtering false positive edges through statistical testing.

- **Combined efficiency**: OCMB requires only 1,980 CI tests compared to IAMB's 5,699 (65% reduction), demonstrating that ordering constraints also improve computational efficiency.

## J. Statistical Query Budget Distribution

Figure 9 visualizes the distribution of query consumption across seeds. OCMB's distribution is remarkably compact (CV=2.1%), with all seeds clustering tightly around 15k–16k queries. IAMB shows similar stability (CV=2.2%) at ∼23k queries. In stark contrast, PC exhibits a heavy-tailed distribution spanning three orders of magnitude, with 2 out of 5 seeds hitting computational limits at 10M queries. This reveals that ordering-constrained discovery is not only more efficient but also more predictable than unconstrained methods. PC's extreme variability (CV=100.9%) means its computational cost is essentially unpredictable in practice, with important implications for production deployment where resource budgets must be planned in advance.

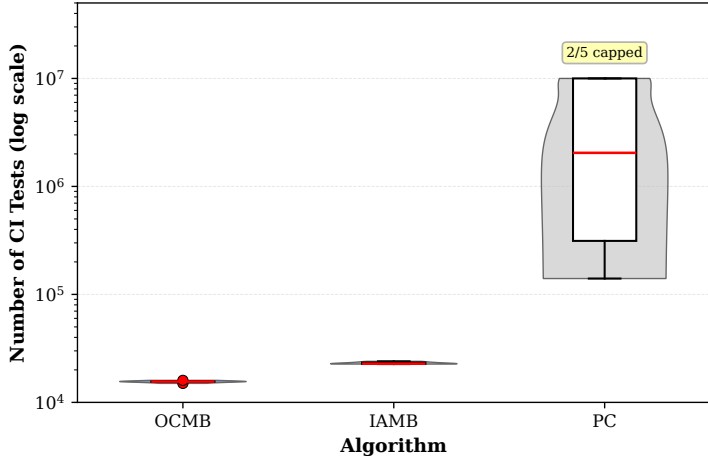

*Figure 9.* Distribution of statistical query consumption across algorithms. Violin plot with overlaid boxplot showing query counts over 5 random seeds on scale-free graphs ($d = 100$, $n = 2000$). Algorithms are ordered by increasing query usage. OCMB shows remarkably stable query consumption (CV=2.1%), while PC exhibits extreme variability (CV=100.9%) with 2/5 runs hitting the 10M computational limit.

## K. Scalability on Scale-Free Graphs

### K.1. Runtime Scaling

Table 14 shows runtime scaling on scale-free graphs with $n = 1000$ samples over 20 random seeds. OCMB scales linearly up to $d = 200$ under a fixed $K/d = 0.1$ ratio. In contrast, PC becomes increasingly expensive and may time out due to large conditioning sets around hubs, with success rates dropping from 100% at $d = 50$ to 40% at $d = 200$.

*Table 14.* Runtime scaling on **scale-free** graphs ($n$=1000). PC time is averaged over successful runs; success rate shown in parentheses.

| $d$ | OCMB Time (s) | OCMB #CI Tests | PC Time (s) |
|-----|---------------|----------------|-------------|
| 20 | 0.8 | 215 | 0.5 (1.0) |
| 50 | 5.9 | 1928 | 59.7 (1.0) |
| 100 | 30.9 | 10159 | 154.9 (0.8) |
| 200 | 134.0 | 51649 | 776.2 (0.4) |

### K.2. Large-Scale Results at $d = 500$

To probe larger scales, we additionally evaluate on $d = 500$ synthetic graphs ($n$=2000). Table 15 shows results on scale-free graphs. In the single successful seed (1/5), OCMB achieves an F1 of 0.541 and SHD of 328, while the ordering backbone alone (CaPS-alone) produces F1 of 0.047 and SHD of 12,361. This preliminary result suggests that local CI refinement can substantially reduce false positives from ordering-only methods, though the low success rate (1/5 SF, 0/5 ER; Section M) precludes strong scaling conclusions. PC ran out of memory in 4/5 runs.

*Table 15.* Preliminary results on $d$=500 **scale-free** graphs ($n$=2000, 1/5 seeds successful). In the successful seed, OCMB substantially reduces SHD compared to the ordering backbone alone.

| Method | F1↑ | SHD↓ | Time (s) | #CI Tests |
|--------|-----|------|----------|-----------|
| **OCMB(caps)** | **0.541** | **328** | 3245 | 373,181 |
| CaPS-alone | 0.047 | 12,361 | 137 | – |

## L. Scalability on Erdős–Rényi Graphs

On ER graphs with homogeneous degree distributions, runtimes increase more mildly for PC compared to scale-free graphs, as typical conditioning sets remain smaller. OCMB scales similarly under fixed $K/d = 0.1$ ratio. Table 16 shows that both

methods scale gracefully on ER graphs, with OCMB maintaining near-linear growth.

*Table 16.* Runtime scaling on **Erdős–Rényi** graphs ($n = 1000$, OCMB uses $K/d = 0.1$).

| $d$ | OCMB Time (s) | OCMB #CI Tests | PC Time (s) |
|---|---|---|---|
| 20 | 0.9 | 231 | 0.2 |
| 50 | 8.5 | 2308 | 1.4 |
| 100 | 39.6 | 12257 | 7.9 |
| 200 | 200.0 | 64381 | 52.1 |

## M. Run Status for $d = 500$ Experiments

At $d = 500$, most methods encounter severe computational challenges. OCMB succeeded in 1/5 seeds on scale-free graphs, while PC ran out of memory in 4/5 runs and MMHC timed out in all runs, reflecting the difficulty of classical constraint-based and hybrid baselines at this scale. The remaining OCMB runs encountered exceptions or timeouts in the current kNN-CMI pipeline rather than graph-level out-of-memory failures. Table 17 provides the complete breakdown of run outcomes across all methods and graph types.

**Engineering Bottleneck.** The dominant computational bottleneck at $d=500$ is the exact kNN-based CMI computation. Each CMI query performs repeated all-pairs $L_\infty$ distance scans to determine the $k$-th neighbor radius, followed by additional radius-based neighbor counts in the $(X, Z)$, $(Y, Z)$, and $Z$ marginal subspaces. As the conditioning set $S$ grows, each distance evaluation touches more coordinates, compounding the cost. The failures at this scale are primarily timeouts in the kNN-CMI pipeline rather than out-of-memory errors. Natural engineering improvements include approximate nearest-neighbor search, batched/cached distance computation, and GPU-accelerated estimators.

*Table 17.* Status summary for $d=500$ synthetic graphs ($n=2000$, 5 seeds).

| Graph | Method | Success | Timeout | Exception | OOM |
|---|---|---|---|---|---|
| | OCMB(caps) | 0 | 2 | 3 | 0 |
| | PC | 0 | 0 | 1 | 4 |
| ER | IAMB | 0 | 2 | 3 | 0 |
| | MMHC | 0 | 5 | 0 | 0 |
| | CaPS-alone | 5 | 0 | 0 | 0 |
| | OCMB(caps) | 1 | 1 | 3 | 0 |
| | PC | 0 | 0 | 1 | 4 |
| SF | IAMB | 0 | 2 | 3 | 0 |
| | MMHC | 0 | 5 | 0 | 0 |
| | CaPS-alone | 5 | 0 | 0 | 0 |

## N. Extended Ordering Error Sensitivity Analysis

This section extends the ordering robustness analysis from Section 4.9 with additional experiments across different $K$ values.

**Analysis Across $K$ Values.** We further compare different candidate set sizes ($K$). Figure 10 shows F1, Parent Recall (covPa), and SHD as functions of the ordering error rate for $K \in \{1, 5, 10\}$ on scale-free graphs with $d=100$ nodes.

**Key Findings.** Table 18 summarizes the results across different $K$ values:

1. **Smaller $K$ is dramatically better**: With $K=1$, OCMB achieves F1=0.84 and SHD=27 at 0% error, compared to F1=0.19 and SHD=695 for $K=10$. This confirms that accurate orderings enable aggressive candidate restriction.

2. **Relative degradation is consistent**: All $K$ values show approximately 33–36% F1 drop from 0% to 50% error, indicating that the relative sensitivity to ordering errors is independent of $K$.

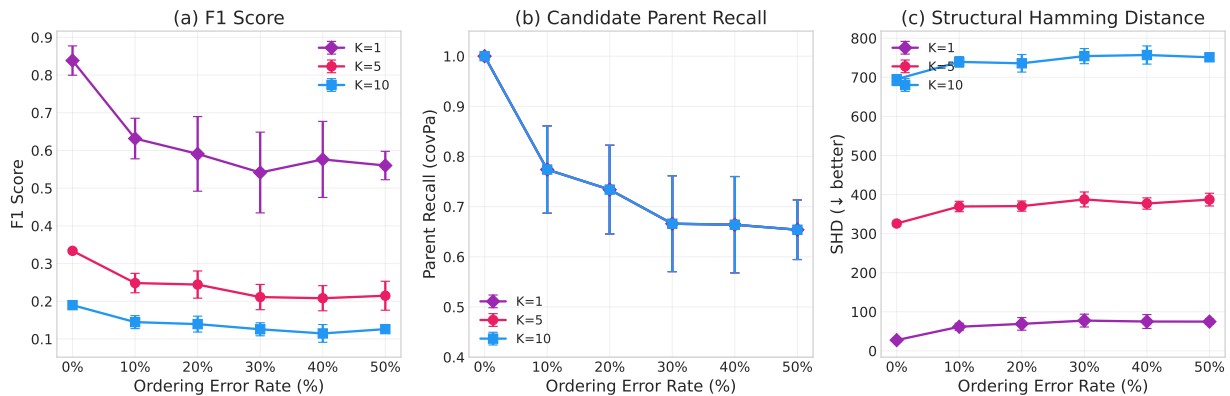

*Figure 10.* OCMB performance under ordering corruption for different $K$ values ($d$=100, $n$=1000). (a) F1 Score, (b) Candidate Parent Recall, (c) Structural Hamming Distance. Smaller $K$ values achieve dramatically better accuracy but are more sensitive to ordering errors in absolute terms.

*Table 18.* Ordering error sensitivity across $K$ values. F1 and SHD at 0% and 50% error rates.

| $K$ | F1@0% | F1@50% | Drop% | SHD@0% | SHD@50% |
|---|---|---|---|---|---|
| 1 | 0.838 | 0.560 | 33.2% | 27.4 | 73.0 |
| 5 | 0.334 | 0.215 | 35.7% | 325.8 | 403.2 |
| 10 | 0.190 | 0.126 | 33.5% | 695.2 | 835.4 |

3. **Parent Recall explains the mechanism**: At 50% error, Parent Recall drops to 0.65 (for $K$=1), meaning 35% of true parents are excluded from candidate sets due to incorrect ordering. This directly bounds recovery capability.

4. **Practical recommendation**: For real-world applications where ordering quality is uncertain, $K$=5 ($K/d$=0.05) provides a balanced trade-off between efficiency and robustness.

## O. Ordering Variants and Random Ordering Ablations

Table 19 presents complete results for all OCMB ordering variants on DREAM4 and Alarm benchmarks, including Oracle and Random orderings. OCMB-oracle achieves the best performance with mean SHD of 208 across DREAM4 networks (ranging from 179 to 251), demonstrating that better ordering quality leads to improved causal discovery accuracy. OCMB-caps (mean SHD: 221) and OCMB-scino (mean SHD: 235) with learned orderings substantially outperform constraint-based baselines PC (mean SHD: 327) and continuous optimization methods like NOTEARS (mean SHD: 794). OCMB-random (mean SHD: 226) yields nontrivial SHD on these benchmarks, which we interpret as a finite-sample regularization effect from bounded candidate restriction rather than evidence that the exact-recovery theory applies under poor orderings. On Alarm, OCMB-oracle achieves SHD of 30, while OCMB-scino achieves SHD of 44, both outperforming PC (53) and NOTEARS (47).

*Table 19.* Complete results on **DREAM4** (100 nodes) and **Alarm** (37 nodes) including all ordering variants.

| Method | Net1 | Net2 | Net3 | Net4 | Net5 | Alarm |
|---|---|---|---|---|---|---|
| PC | 305 | 385 | 305 | 332 | 307 | 53 |
| NOTEARS | 704 | 913 | 691 | 868 | 793 | 47 |
| **OCMB-oracle** | **179** | **251** | **198** | **219** | **195** | **30** |
| OCMB-caps | 187 | 263 | 214 | 229 | 213 | 49 |
| OCMB-scino | 200 | 286 | 210 | 253 | 227 | 44 |
| OCMB-random | 194 | 274 | 216 | 230 | 217 | 51 |

**Random Ordering Mechanism Analysis.** Even when Assumption 3.3 is violated (as in OCMB-random), bounded candidate restriction can still yield nontrivial finite-sample behavior on some benchmarks. In classical constraint-based

methods like PC, hub nodes require conditioning on large neighborhoods ($O(\Delta)$ where $\Delta$ is the node degree), leading to statistical unreliability. By contrast, OCMB conditions only on the candidate set $\mathcal{C}_{\mathrm{MB}}(X_i)$ whose size is bounded by the sparsity parameter $K$. This "regularization by candidate restriction" can improve finite-sample CI test reliability even when the ordering provides little causal information. However, random-ordering results should not be interpreted as robustness to arbitrary poor orderings: the controlled decomposition in Table 7 shows that random orderings can collapse F1 when parent recall is low. Overall, OCMB's gains stem from two complementary sources: (1) correct ordering provides parent recall guarantees (Theorem 3.6), and (2) candidate set restriction improves statistical stability when the resulting candidate sets still retain useful signal.

