# OpenReview forum: "Global Directional Priors with Local Statistical Validation for Scalable Causal Discovery"
_ICML.cc/2026/Conference — ICML 2026 regular_

### Official Review · Reviewer_sWgb · 2026-03-04

**Soundness:** 3
**Presentation:** 2
**Significance:** 3
**Originality:** 2
**Overall Recommendation:** 4
**Confidence:** 3

**Summary:**

The central question discussed by the paper is how to maintain the statistical reliability and scalability of constraint-based causal discovery in high-dimensional, hub-dominated networks, where traditional conditional independence (CI) tests degrade rapidly due to uncontrollably large conditioning sets. This submission's fundamental contribution pertains to the development of Ordering-Constrained Markov Blanket discovery (OCMB), a two-stage paradigm that explicitly treats conditioning-set dimensionality as a primary structural constraint. Instead of directly discovering an undirected skeleton or globally optimizing a structure score, OCMB first utilizes a lightweight global ordering estimator to establish a directional prior. This prior filters the search space, restricting candidate parents to a strictly bounded set of topological predecessors. In the second stage, local statistical validation is executed entirely within these low-dimensional candidate sets. The authors provide theoretical guarantees for parent and Markov blanket recovery under a soft "high-recall ordering" assumption, mathematically showing that the global ordering acts as a screening mechanism and does not need to be perfectly accurate. Empirically, the study demonstrates that OCMB drastically reduces the number of required CI queries—often by orders of magnitude—while significantly improving structural accuracy metrics on scale-free graphs and real-world biological benchmarks compared to classical constraint-based and hybrid methods.

**Compliance With Llm Reviewing Policy:**

Affirmed.

**Key Questions For Authors:**

1. Table 7 shows that the $Rec_{Pa}$ for OCMB-caps is 0.57, meaning 43% of the true parents are excluded from the candidate sets. Since OCMB currently lacks a mechanism to recover these edges, have you considered introducing a "fallback" or post-processing phase to rescue missed parents based on the Markov Blankets identified in Stage II?

2. Appendix Table 8 reveals that OCMB's directed F1 score on ER graphs (0.236) falls significantly behind PC (0.663). How does this vulnerability affect the overall applicability of OCMB, particularly on non-hub-dominated graphs, and why was this structural limitation not openly discussed in the main text?

3. Regarding the $d=500$ scalability experiments, Table 17 indicates that OCMB succeeded in only 1/5 seeds on SF graphs and 0/5 on ER graphs. What is the primary bottleneck causing these failures (e.g., Out-Of-Memory vs. timeouts)? Are there straightforward engineering optimizations that could resolve this?

4. The method employs a fixed CI threshold of $\alpha=0.01$. Given that estimated CMI statistics often scale with the sample size $n$ and the conditioning set dimension, does the optimal $\alpha$ shift significantly under different $(n, d)$ settings? Have you conducted a sensitivity analysis for $\alpha$?

5. Backbones like CaPS and SciNO rely heavily on the Additive Noise Model assumption. How does OCMB perform when the underlying data-generating process violates this assumption (e.g., multiplicative noise or discrete data)? Do you have empirical results or theoretical insights regarding this scenario?

**Limitations:**

1. The recommended scenarios listed in Appendix G are somewhat heuristic. The concession that "PC may remain competitive" on homogeneous networks is too vague. A more rigorous empirical formalization is needed to establish the exact boundaries where OCMB loses its advantage.

2. The paper emphasizes scenarios where OCMB performs well or degrades gracefully under synthetic ordering noise (Figure 6). It lacks a deep dive into regimes where the framework might fail catastrophically. For instance, analyzing how OCMB behaves when the initial backbone completely collapses would provide a more balanced perspective for practitioners.

**Strengths And Weaknesses:**

### **Strengths**

1.The paper accurately targets a critical bottleneck in constraint-based causal discovery: the degradation of conditional independence (CI) tests due to large conditioning sets in hub-dominated networks. Elevating "conditioning-set dimensionality" to a first-class design constraint is a highly practical and somewhat overlooked perspective.

2.The two-stage design of OCMB is highly modular. Table 3 effectively demonstrates that both the Stage I backbone and the Stage II local learner can be flexibly interchanged, confirming that the performance gains stem from the overarching framework rather than specific components.

3.The ablation study in Table 7 is particularly strong. By systematically evaluating five variants, it successfully disentangles the contributions of ordering quality and score informativeness. This prevents readers from wrongly attributing the performance gains solely to sparsity constraints.

4.The method is rigorously evaluated on standard benchmarks. Notably, it reduces the SHD by 43-56% on DREAM3 compared to EEMBI. Furthermore, on scale-free graphs, OCMB achieves competitive or better accuracy than PC while requiring roughly 5,000 times fewer CI tests, which is a massive efficiency leap.

5.The paper is well-written. Figures 1 and 2 excellently illustrate the algorithmic pipeline and intuitively clarify the fundamental difference between OCMB and existing hybrid methods like MMHC.

### **Weaknesses**

1. The theoretical contributions are mostly formalizations of the algorithmic steps rather than profound insights. Theorem 3.6 is essentially a direct restatement of Assumptions 3.3 and 3.5. Similarly, Theorem 3.8 is a straightforward application of standard IAMB consistency.

2. The theoretical guarantees heavily rely on the "High-Recall Ordering" assumptions (3.3 and 3.5), which assume the backbone places true parents in the top-$K$ list. However, Table 7 shows the $Rec_{Pa}$ for OCMB-caps is only 0.57. This indicates that the assumptions are frequently violated in practice, significantly discounting the practical value of the theoretical claims.

3. OCMB struggles significantly with directed edge recovery on Erdos-Renyi graphs. As shown in Appendix Table 8, OCMB's directed F1 is only 0.236, falling far behind PC and GRaSP. This exposes a fundamental flaw: in non-hub-dominated graphs, ordering priors can introduce severe false negatives. Reporting only skeleton metrics for ER graphs in the main text (Table 2) selectively obscures this weakness.

4. The claims regarding scalability up to 500 nodes lack statistical significance. According to Appendix Table 17, OCMB only succeeded in 1 out of 5 seeds on SF graphs and failed entirely (0/5) on ER graphs. Presenting the $d=500$ results in the main text (Table 15) based on a single successful run is misleading.

5. Minor Presentation Issues:

   **Page 3:** Assumption 3.3 uses the notation $Pred_i$ before it is formally defined in Assumption 3.5. It should be defined upon first use.

   **Page 3:** The statement of Theorem 3.6 uses deterministic language ("for all $i$"), but the corresponding proof sketch relies on a probabilistic bound ($1 - \delta$). The theorem statement should accurately reflect this probabilistic nature.

   **Page 6:** The caption of Figure 3 mentions "$n=2800$ tests", which is confusing given the standard sample size is $n=1000$ throughout the text. Please clarify whether this refers to the number of CI tests or a different sample size.

---

> ### Author Rebuttal · Authors · 2026-03-26
>
> We thank the reviewer for the constructive and technically sharp feedback. We agree that the paper should state its failure regimes more explicitly, and we address the main points below.
>
> ### (1) Low parent recall and a possible fallback phase
>
> You are correct that the current theory is conditional on Stage-I recall, and that in practice the learned backbone can miss true parents. This is already visible in **Table 7**, where `OCMB-caps` has candidate parent recall $\mathrm{Rec}_{\mathrm{Pa}}=0.57$. In the current theory, this enters directly through **Theorem 3.6**: once a true parent is excluded from the Stage-I candidate set, exact parent recovery is no longer covered by the guarantee. At present, **OCMB has no rescue mechanism for such missed parents**. This is a real limitation of the current method.
>
> We agree that a lightweight fallback or post-processing phase is a promising extension, but we do not want to overstate it as something the current method already provides.
>
> Empirically, imperfect Stage-I recall does not necessarily imply catastrophic collapse: the ordering-corruption study in **Figure 6** shows graceful degradation, suggesting that bounded candidate restriction can still stabilize downstream CI testing in finite samples even when the prior is imperfect. We stress, however, that this is an **empirical finite-sample effect**, not something covered by the exact-recovery theorem.
>
> ### (2) Directed F1 on ER graphs
>
> We agree that the ER directed-edge results expose an important structural boundary of OCMB. The current paper already shows this split:
>
> - **Table 2** shows that `OCMB-HITON` remains competitive on **skeleton** recovery on ER graphs ($\mathrm{F1}_{\mathrm{skel}}=0.717$, essentially matching PC at $0.716$);
> - **Appendix Table 8** shows that directed recovery is much weaker there ($\mathrm{F1}=0.236$ for `OCMB` and $0.355$ for `OCMB-HITON`, versus $0.663$ for `PC`).
>
> In homogeneous ER graphs, the ordering prior is less informative, so directional restrictions can introduce false negatives. We will move this limitation from the appendix into the main text.
>
> ### (3) $d=500$ scalability evidence
>
> We agree that the current $d=500$ evidence is statistically weak and should not be presented too strongly. The appendix shows two distinct facts:
>
> - **Appendix Table 15** reports the single successful large-scale SF result: `OCMB(caps)` achieves $\mathrm{SHD}=328$, compared with $\mathrm{SHD}=12{,}361$ for `CaPS-alone`;
> - **Appendix Table 17** shows the corresponding run-status weakness: `OCMB(caps)` succeeds on only `1/5` SF seeds and `0/5` ER seeds.
>
> The remaining OCMB runs are mainly **timeouts and current implementation limitations** in the kNN-CMI pipeline, while `PC` mostly fails by **OOM** in the same table. We will therefore tone down the scalability claim and present the `d=500` result as preliminary evidence of feasibility rather than a strong scaling conclusion.
>
> ### (4) Sensitivity to $\alpha$
>
> We agree with this concern. Because $\alpha$ is applied to an estimated CMI statistic rather than to a calibrated $p$-value, its effective scale can shift with sample size and conditioning dimension. We have **not** yet performed a dedicated $\alpha$ sweep; in the current paper this appears only as a stated limitation in **Appendix F, Point 2**. We will make that limitation more explicit in the main text rather than leaving it only in the appendix.
>
> ### (5) Backbone assumptions and ANM violations
>
> We do not claim robustness to arbitrary violations of the backbone assumptions, and we do not currently present a dedicated multiplicative-noise or discrete-data stress test. The theory itself is conditional: **Theorem 3.6 / Theorem 3.8** require sufficient Stage-I recall and MB-superset coverage, but do not directly prove robustness to ANM violations. On the empirical side, the benchmark results on **Sachs, DREAM3/4, and Alarm** (**Tables 4-6**) suggest that OCMB can still be practically useful beyond the main synthetic nonlinear-Gaussian setup, but we agree that this does **not** substitute for a controlled stress test under explicit ANM violations. The closest controlled evidence is the degraded-prior analysis in **Table 7** and **Appendix Table 19**, which supports only a narrower point: bounded candidate restriction can still provide some finite-sample regularization benefit when Stage-I quality is imperfect. We will revise the text to make that boundary explicit.
>
> ### (6) Minor presentation issues
>
> We will also incorporate the presentation fixes you identified: define $\mathrm{Pred}_i$ before first use, state the probabilistic nature of the theorem more accurately, and clarify that “$n=2800$ tests” in **Figure 3** refers to the number of CI queries, not the dataset sample size.

---

> > ### Author Rebuttal · Reviewer_sWgb · 2026-04-03
> >
> > I appreciate your candid acknowledgement of OCMB's structural boundaries and your commitment to making these limitations explicitly clear in the main text. I am satisfied with your responses to my primary concerns and intend to maintain my score.
> >
> > I have two minor follow-up questions based strictly on your rebuttal, aimed at providing better practical guidance for future readers:
> >
> > 1.You clarified that the failures at the $d=500$ scale are primarily due to "timeouts and current implementation limitations in the kNN-CMI pipeline" rather than Out-Of-Memory (OOM) errors. Out of curiosity for the engineering aspects of this framework, could you briefly elaborate on what specific operation within this pipeline constitutes the primary computational bottleneck? Is it the nearest neighbor search in high dimensions, the distance calculations, or something else?
> >
> > 2.You agreed that the effective scale of the fixed $\alpha = 0.01$ shifts with sample size ($n$) and conditioning dimension ($d$), and noted that a dedicated sweep was not performed. Given this, how would you recommend future practitioners approach setting or tuning this $\alpha$ threshold when applying OCMB to entirely new, real-world datasets with completely different $(n, d)$ characteristics?

---

> > > ### Author Response · Authors · 2026-04-04
> > >
> > > We greatly appreciate your thoughtful follow-up questions and your continued engagement with our work. We are glad that our previous response addressed your main concerns. Below we provide brief answers to the two practical points you raised, and we will clarify them in the final version.
> > >
> > > ### 1. Engineering bottleneck in the kNN-CMI pipeline
> > >
> > > The dominant bottleneck is the exact kNN-based CMI computation itself. In our current Numba-CUDA implementation, each CMI query first performs repeated all-pairs $L_\infty$ distance scans to determine the $k$-NN radius $\rho$, and then performs additional radius-based neighbor counts in the $(X,Z)$, $(Y,Z)$, and $Z$ subspaces. Thus, the main cost is the combination of:
> > >
> > > 1. **Exact nearest-neighbor search:** for each sample, the kernel scans all other samples to identify the $k$-th neighbor radius.
> > > 2. **Repeated neighbor counting:** after $\rho$ is determined, the estimator still needs additional passes to count neighbors within that radius in the marginal subspaces.
> > >
> > > As the conditioning set $S$ grows, each distance evaluation touches more coordinates, and the resulting neighbor counting also becomes slower. So the main issue is not a separate post-processing step, but the cost of exact nearest-neighbor search plus repeated counting inside every CMI query. This is why, at the $d=500$ scale, the OCMB failures are mainly timeouts and runtime limitations in the current kNN-CMI pipeline rather than graph-level OOM. Natural future engineering improvements would include better batching/caching and approximate or indexed nearest-neighbor search.
> > >
> > > ### 2. Practical guidance for setting $\alpha$ on new datasets
> > >
> > > We agree this is an important practical issue. In our setup, $\alpha$ is a threshold on an estimated CMI statistic, not a calibrated $p$-value, so its effective scale can change with sample size $n$ and conditioning dimension $|S|$.
> > >
> > > For practitioners applying OCMB to a new dataset, our practical recommendation is:
> > >
> > > 1. **Start from the default used in the paper** ($\alpha = 0.01$).
> > > 2. **Run a small sensitivity sweep** (for example, $\{0.005, 0.01, 0.02\}$, and optionally $0.05$) and check whether the learned edges are stable across nearby choices.
> > > 3. **If computation allows, use permutation-based calibration** to estimate a null scale for the target sample size and conditioning dimensions more systematically.
> > >
> > > As a simple rule of thumb, when $n$ is smaller or the conditioning sets are larger, the CMI estimates are typically less stable, so we would rely less on a single fixed $\alpha$ and more on sensitivity checks or permutation-based calibration. We will clarify this practical recommendation in the final version.

---

### Official Review · Reviewer_bvAH · 2026-03-05

**Soundness:** 3
**Presentation:** 3
**Significance:** 3
**Originality:** 3
**Overall Recommendation:** 5
**Confidence:** 3

**Summary:**

This work proposes OCMB, a two-stage causal discovery framework that overcomes the statistical failure of traditional CI-based methods in hub-dominated graphs. The idea is to use a learned topological ordering as a directional prior to construct small candidate sets, then perform CI tests only within these bounded sets. This ensures all tests operate under low-dimensional conditions and avoids the "error cascade" that plagues methods like PC. From a Bayesian perspective, Stage I provides a structured prior that narrows the hypothesis space; Stage II updates this prior locally using CI tests as likelihood evidence. Experiments on synthetic and real benchmarks show OCMB achieves state-of-the-art accuracy with 2-3 orders of magnitude fewer CI tests than PC-family methods. Theoretical guarantees and ablation studies confirm that both directional informativeness and dimensionality control drive these gains.

**Compliance With Llm Reviewing Policy:**

Affirmed.

**Final Justification:**

It addresses most of my concerns.

**Key Questions For Authors:**

1. Figure 1 is confusing. Given the topological ordering $\pi$ in the INPUT panel (where node 6 comes before node 4), how do we end up with an edge $4 \rightarrow 6$ in the DAG? That direction seems to contradict the ordering. Could you clarify how the final graph is constructed from the ordering?

2. The proposed method under the random ordering setting appears to perform surprisingly well. Table 19 shows that OCMB with *random* ordering and random scores performs almost as well as the learned orderings on DREAM4 (SHD 226 vs. 221/235). This seems to directly contradict Lemma 3.7, which assumes a high-recall ordering. The work mentions "regularization by candidate restriction" as an explanation. But if the candidate sets don't contain the true parents, why doesn't the method fail catastrophically? Does this mean the theoretical guarantees are *sufficient* but not *necessary*, and that OCMB's gains actually come from controlling conditioning set size than from getting the ordering right?

3. Are spouses really redundant? The work argues that under an ordering prior, spouses become redundant for Markov blanket identification because directionality is already resolved. But the classical Markov blanket includes spouses to ensure conditional independence with all other nodes. Can you formally justify that spouses are unnecessary under an ordering prior? If they're truly redundant, why not drop them entirely from Stage II to reduce even more number of CI tests? And if you keep them, does that mean they still matter, especially when the ordering prior is noisy?

**Limitations:**

Yes

**Strengths And Weaknesses:**

### Strengths
1. The problem is well-motivated problem, as conditioning-set dimensionality is a main cause of CI-based methods' failure in hub graphs.
2. The presentation is clear, well-written, and self-contained, with comprehensive experiments on synthetic and real-world benchmarks.
3. The work has elegant method design, which decouples global ordering (directional prior) from local CI tests, ensuring bounded conditioning sets and avoiding statistical error cascade.

### Weaknesses
The theoretical contributions are modest. While the work provides several formal theoretical results (Candidate Parent Recall, Consistent MB Recovery), these are established under strong assumptions (High-Recall Ordering, Top-K Parent Recall) that may be difficult to verify in practice.

There are a few minor issues:
1. Assumption 3.3 introduces $Pred_i$ without defining it.
2. Theorem 3.8 states "Under Assumption 3.2 and Lemma 3.7," why the result is imposed as part of the assumption?
3. Figure 1 is provided but its workflow is not explained in detail; Figure 2 is not cited in the main text.
4. Table 1 lists "OCMB-HITON" without explaining what it represents.
5. Figure 4 uses colors that are difficult to distinguish.

---

> ### Author Rebuttal · Authors · 2026-03-25
>
> We thank the reviewer for the careful reading. Your questions go directly to the theory/empirics interface, and we address them below.
>
> ### (1) Random ordering versus Lemma 3.7
>
> We agree that this should be separated more clearly. In our paper, **Theorem 3.6** gives the candidate-parent recall guarantee under the high-recall assumptions, and **Lemma 3.7** extends this to MB-superset coverage. These are **sufficient** conditions for exact recovery; they are not necessary conditions for observing any empirical benefit.
>
> The paper contains two different settings:
>
> - In the **controlled synthetic decomposition** (**Table 7**), random ordering performs very poorly: `OCMB-random-ord` has $\mathrm{F1}=0.027$, compared with $\mathrm{F1}=0.437$ for `OCMB-caps` and $\mathrm{F1}=0.834$ for `OCMB-oracle`. This is fully consistent with the theorem’s premise being violated.
> - In the **DREAM4/Alarm ordering-variant appendix** (**Appendix Table 19**), random ordering still yields nontrivial benchmark SHD; the appendix reports mean $\mathrm{SHD}=226$ for `OCMB-random`, compared with $\mathrm{SHD}=221$ for `OCMB-caps` and $\mathrm{SHD}=235$ for `OCMB-scino`.
>
> We interpret the second observation only as evidence that bounded candidate restriction can still provide some finite-sample regularization in certain benchmark settings, not as evidence that the theorem remains applicable there. We will revise the paper to make this distinction explicit.
>
> ### (2) Are spouses redundant?
>
> We agree that the text should distinguish **theoretical sufficiency** from **practical necessity** more clearly.
>
> For the theoretical MB-superset result, **Lemma 3.7** uses spouse closure as a sufficient construction. In practical DAG recovery under an ordering prior, however, orientation is already supplied by Stage I, so spouse closure can be empirically less important. This is exactly what **Appendix Table 12** shows: disabling spouse closure reduces CI tests from $16{,}259$ to $7{,}400$, while skeleton $\mathrm{F1}$ changes only from $0.431$ to $0.445$.
>
> We will revise the manuscript to say this more carefully: spouse closure is useful for the theorem as stated, but not always empirically necessary for the practical OCMB pipeline. We therefore do **not** claim spouses are universally redundant; rather, we keep spouse closure as a conservative option to preserve MB coverage under the theorem, even if cleaner settings do not always require it.
>
> ### (3) Figure 1 contradiction
>
> You are right that **Figure 1** is inconsistent with the ordering constraint. The final DAG should respect $\pi(j) < \pi(i)$ for every edge $X_j \to X_i$, and we will correct the figure accordingly.
>
> ### (4) Minor presentation issues
>
> We will also revise the presentation by defining $\mathrm{Pred}_i$ before first use, restating **Theorem 3.8** more accurately (since **Lemma 3.7** is a derived sufficient result, not an assumption), citing/explaining Figure 2 more clearly, defining `OCMB-HITON` on first use as the OCMB variant with HITON as the Stage-II MB learner, and improving the Figure 4 palette.

---

> > ### Author Rebuttal · Reviewer_bvAH · 2026-04-01
> >
> > The rebuttal argues that spouse closure is theoretically sufficient for ensuring Markov blanket coverage, yet empirically appears less critical when a high-quality ordering prior is available. This raises the question of how the method behaves when the ordering quality is poor, such as in the OCMB-random setting. In such cases, would omitting spouse closure lead to a considerable performance degradation?
> >
> > More importantly, if the theoretically sufficient condition can be relaxed in practice without adversely affecting results, what is the source of this discrepancy? Does this mean that the theoretical condition is stronger than necessary, or that the empirical evaluations do not adequately capture scenarios in which spouse closure becomes essential?

---

> > > ### Author Response · Authors · 2026-04-02
> > >
> > > We thank the reviewer for this precise follow-up. The question identifies a genuine theory–practice gap that deserves a clear explanation.
> > >
> > > ---
> > >
> > > ### The source of the discrepancy
> > >
> > > The discrepancy arises because **Lemma 3.7 and the practical OCMB pipeline solve different problems**, and spouse closure is needed for one but not the other.
> > >
> > > **Lemma 3.7** guarantees that the candidate set $\mathcal{C}\_{\mathrm{MB}}(X\_i)$ is a *superset* of the true Markov blanket $\mathrm{MB}\_{\mathcal{G}^{*}}(X\_i)$. This is a strict statement about **undirected neighborhood recovery**: to identify all variables that are statistically relevant to $X\_i$, one must include spouses (co-parents of children) because they are part of the classical Markov blanket by definition. Without spouse closure, the candidate set violates this theoretical superset guarantee.
> > >
> > > **The practical OCMB pipeline (Algorithm 1)**, however, does not output the full Markov blanket. It recovers a **directed parent set**: the final output (line 10) constructs edges as $A\_{j,i} \leftarrow 1$ strictly for $j \in \widehat{\mathrm{MB}}\_i \cap \mathcal{C}\_{\mathrm{Pa}}(X\_i)$. That is, only variables that are both in the recovered MB *and* in the candidate parent set become edges. Spouses, by definition, are neither parents nor children, so **spouses never directly become edges in the output DAG**.
> > >
> > > In classical unconstrained MB-based methods, spouses serve two indirect but crucial roles: (1) as conditioning variables to block spurious paths opened by colliders (d-separation correctness), and (2) to resolve V-structures for edge orientation. In OCMB, the ordering prior already resolves edge directions: if $\pi(j) < \pi(i)$, $X\_j$ is a candidate parent of $X\_i$, making role (2) redundant. For role (1), the bounded candidate sets under a good ordering ($|\mathcal{C}\_{\mathrm{MB}}| \leq 5$, Table 7) limit the number of active paths that need blocking, reducing the practical need for spouse-based conditioning.
> > >
> > > ---
> > >
> > > ### Does ordering quality affect spouse importance?
> > >
> > > This addresses your first question. We reason that **spouse closure appears empirically largely dispensable across both regimes**:
> > >
> > > * **When the ordering is high-quality** (e.g., OCMB-oracle/caps): The candidate parent sets already contain the true parents with a small $|\mathcal{C}\_{\mathrm{Pa}}|$. Bounded candidate sets naturally prevent spurious path activations. Adding spouses is unnecessary because the ordering provides direction and the small conditioning sets are sufficient for accurate CI tests. This is exactly what Appendix Table 12 shows (spouse=off: F1=0.445 vs. spouse=on: F1=0.431).
> > >
> > > * **When the ordering is poor** (e.g., OCMB-random): The candidate parent sets have low recall ($\mathrm{Rec}\_{\mathrm{Pa}}=0.14$, Table 7), meaning 86% of true parents are excluded by the ordering. In this regime, the method is fundamentally bottlenecked by **false negatives**---missing true parents that no amount of conditioning-set refinement can recover. The candidate set is already large (Table 7: median $|\mathcal{C}\_{\mathrm{MB}}|=55.5$), and performance collapses to F1=0.027. **The dominant error source is irrecoverable parent exclusion, not conditioning-set incompleteness.** Whether or not spouses are included in the conditioning set, the 86% of missing parents remain missing---spouse closure addresses conditioning quality, but cannot compensate for the fundamental recall deficit imposed by the ordering. **This also explains why, in practice, removing spouse closure in such regimes is not expected to materially change outcomes, as the primary failure mode lies upstream in the ordering stage.**
> > >
> > > In other words: **spouse closure addresses a secondary concern (classical MB completeness) that is rendered redundant by the ordering prior in the good regime, and eclipsed by parent recall errors in the poor regime.**
> > >
> > > ---
> > >
> > > ### Is the theoretical condition stronger than necessary?
> > >
> > > This answers your second question: Yes, the condition in Lemma 3.7 is intentionally **sufficient but not tight**.
> > >
> > > Lemma 3.7 provides a **conservative sufficient condition** to prove that OCMB *can* act as a strict undirected Markov Blanket estimator if desired. However, the practical pipeline targets a narrower goal—directed parent set recovery under ordering constraints—for which the full MB-superset guarantee is stronger than strictly needed.
> > >
> > > We will revise the manuscript to make this distinction explicit: Lemma 3.7 provides the sufficient condition for classical MB consistency, while the edge construction in Algorithm 1 structurally relaxes the empirical need for spouse closure. **Importantly, this gap does not undermine OCMB's correctness; it reflects that the theoretical condition is conservative relative to the practical objective, and does not limit OCMB's empirical robustness or validity.**

---

### Official Review · Reviewer_4u24 · 2026-03-05

**Soundness:** 1
**Presentation:** 1
**Significance:** 2
**Originality:** 2
**Overall Recommendation:** 4
**Confidence:** 3

**Summary:**

This paper proposes an ordering-based causal discovery algorithm that first finds the causal order using existing algorithms, and then runs a modified version of existing Markov-blanket based discovery algorithms to obtain the full graph. In particular, the ordering allows for restricting the parents and children sets of a variable, which ultimately restricts its candidate Markov blanket, which hence requires fewer CI tests in the Markov-blanket based discovery phase. The authors provide a complexity analysis and an extensive experimental evaluation.

**Compliance With Llm Reviewing Policy:**

Affirmed.

**Final Justification:**

The rebuttal answered some of my concerns. While, I am still not fully sure about the benefits of the trade-off between a ordering-based methods at stage 1 instead of constrained-based methods, w.r.t computational requirements (which is a key issue in causal discovery), assumptions and accuracy, the authors have convinced me that it is a possible approach to mitigate higher-order tests in the scale-free setting. Thus, I'm increasing my score.

**Key Questions For Authors:**

1. What are the computational and statistical complexities of discovering the causal ordering at the beginning of the proposed algorithm?
2. Since several ordering-based algorithms offer strategies to recover the true graph from the discovered orderings, what are the benefits of OCMB compared to an end-to-end ordering-based algorithm?

**Limitations:**

yes

**Strengths And Weaknesses:**

## Soundness

Assumption 3.2 (i) is missing the causal Markov assumption, since faithfulness only ensures that CI in data $\implies$ d-separation, but **not** the other way around.

The main results highlight that OCMB-HITON performs 5000 times fewer CI tests than PC. However, the running time suggests that the majority of the work is performed by the ordering backbone, which uses score-matching instead of CI tests. Then, comparing the CI tests performed by HITON to finish the remaining work to the global learning of PC is **highly misleading**. In particular, the running time suggests that the OCMB variants are more expensive than PC. I am also confused why the authors highlight a running time of 3.6 seconds in bold in Table 1, when both PC and CaPS-alone achieves lower running times.

What are the assumptions of the DAGMA-linear method? In particular, if it assumes a linear structural equation model, which is mis-specified according to the experimental setup, then how is it possible that it achieves the best SHD and F1 scores in Table 1?

I am missing an ablation about utilising different ordering-based methods in the beginning of the algorithm.

## Presentation

The related work is missing literature on constraint-based causal discovery methods that explicitly limit the size of conditioning sets with the same motivation as this work. [Textor et al. 2015] only consider conditioning sets of size 0, [Wienöbst et al. 2020] and [Kocaoglu 2023] consider all CI relations up to a certain size of conditioning set. [Lee et al. 2025] develop a PC variant that considers CI tests where the conditioning set is bounded. The authors should position their paper compared to this line of work.

The paper is also missing a background section on ordering-based methods and Markov-blanket based causal discovery methods. In particular, the paper does not mention anything about the assumptions and computational complexities of ordering-based methods, even though these perform a significant portion of the work in the proposed algorithm.

The proposed algorithm relies on the CaPS algorithm to find the causal orderings. The CaPS algorithm is cited several times, always with the wrong reference of (Rolland et al., 2022), who proposed the SCORE algorithm. The CaPS algorithm was instead proposed by [Xu et al. 2024], which is not cited in the paper.

The paper states that if the true parent set is empty, then the candidate set is also empty. Is this part of Assumption 3.5 or does this follow from the previous assumptions? Why is it important to assume this edge case, when for non-empty true parent sets we only assume a subset, instead of an equality relation.

[Textor et al. 2015] Learning from pairwise marginal independencies. UAI
[Wienöbst et al. 2020] Recovering causal structures from low-order conditional independencies. AAAI
[Kocaoglu 2023] Characterization and learning of causal graphs with small conditioning sets. NeurIPS
[Lee et al. 2025] Constraint-based Causal Discovery from a Collection of Conditioning Sets. UAI
[Xu et al. 2024] Ordering-Based Causal Discovery for Linear and Nonlinear Relations, NeurIPS

## Significance

I find the general motivation of the paper to limit separating set sizes to be relevant and useful. However, the theoretical insights of the paper are somewhat limited and some of the experimental results are confusing.

## Originality

Utilizing causal orderings for causal discovery is a well-explored idea. The idea to use them for Markov-blanket based discovery algorithms is new, but I cannot find the motivation of restricting separating set sizes strong enough when the complexities of finding causal orderings are not discussed.

---

> ### Author Rebuttal · Authors · 2026-03-25
>
> We thank the reviewer for the review. Several points identify genuine problems in the draft, and we will revise accordingly.
>
> ### (1) Assumption 3.2 and CaPS citation
>
> You are right that Assumption 3.2 is imprecise. We stated faithfulness without clearly separating it from the **Causal Markov condition**, and will revise Assumption 3.2 accordingly.
>
> You are also right that CaPS is cited incorrectly. We will replace the current citation with **Xu et al. (2024)** and distinguish it clearly from SCORE / Rolland et al. (2022).
>
> ### (2) CI-test count versus runtime
>
> We agree that the current presentation mixes two notions of efficiency. In **Table 1**, the **CI-test count** reflects the Stage-II statistical query burden, whereas the reported **runtime** is for the full pipeline; placing “5000x fewer CI tests than PC” next to full-pipeline runtime can therefore overstate end-to-end efficiency.
>
> At $d=100$, Table 1 shows that `OCMB-HITON` takes $3.6\,\mathrm{s}$, whereas `PC` takes $1.9\,\mathrm{s}$, yet the same table reports a much smaller Stage-II CI burden ($704$ vs. $3{,}413{,}957$ tests). We will revise the caption/text, remove the bold formatting on runtime, and present CI-test count only as a measure of **local CI burden**, not as a claim of full-pipeline superiority over PC.
>
> Our intended claim is narrower: OCMB trades additional Stage-I computation for smaller candidate sets and lower-dimensional CI tests in Stage II, which is intended to improve finite-sample reliability in hub-heavy regimes.
>
> ### (3) Complexity and assumptions of the ordering stage
>
> We agree that Stage I is under-explained. OCMB is a **framework**, so the computational/statistical assumptions of Stage I are inherited from the chosen backbone (e.g., CaPS or SciNO). Our theory is therefore **conditional**: if the backbone yields candidate parent sets with sufficiently high recall, then the constrained local CI phase is consistent. We will make clear that this is not an ad hoc extra assumption, but the interface condition required when combining a learned structural prior with local CI-based refinement.
>
> We will add a short paragraph clarifying the role of the backbone, that its cost can dominate runtime in small/medium settings, and that OCMB’s guarantees depend on Stage-I candidate-parent recall rather than globally correct ordering.
>
> ### (4) Relation to bounded-conditioning-set literature
>
> Thank you for pointing out this line of work. We will add **Textor et al. (UAI 2015), Wienoebst et al. (AAAI 2020), Kocaoglu (NeurIPS 2023), and Lee et al. (UAI 2025)**.
>
> Our intended distinction is that prior bounded-CI methods impose a **global cap on conditioning-set size**, whereas OCMB uses an ordering prior to restrict the **candidate search space before CI testing**. We will clarify that OCMB is **complementary** to bounded-order CI methods rather than a reimplementation of them.
>
> ### (5) Benefits over end-to-end ordering methods
>
> The practical benefit is that ordering methods are effective at proposing a sparse directional search space, while local CI tests are better suited to pruning false positives than global score thresholding. OCMB therefore uses ordering as a **screening prior** and local CI as a refinement step.
>
> The results already show this benefit:
>
> - in **Table 1** (scale-free, $d=100$, $n=1000$), `CaPS-alone` has $\mathrm{SHD}=446.0$, while `OCMB-HITON` has $\mathrm{SHD}=60.3$;
> - in **Appendix Table 15** (scale-free, $d=500$, $n=2000$), `CaPS-alone` has $\mathrm{SHD}=12{,}361$, while `OCMB(caps)` (OCMB using CaPS as the Stage-I backbone) has $\mathrm{SHD}=328$.
>
> We already include an ordering-stage ablation comparing different Stage-I priors (`CaPS / SciNO / Oracle / Random`) on DREAM4/Alarm in **Appendix Table 19**, and will surface this more clearly in the paper.
>
> ### (6) DAGMA-linear on nonlinear data
>
> We agree that the manuscript does not discuss this carefully enough. We do **not** claim that DAGMA-linear is assumption-matched to our nonlinear SEM setup. We include it as a strong empirical baseline, and its performance should be treated as an empirical observation rather than evidence that its modeling assumptions hold exactly. One plausible explanation is that its sparsity/acyclicity regularization provides a favorable inductive bias on hub-dominated scale-free graphs even under some model mismatch.
>
> ### (7) Empty-parent-set edge case in Assumption 3.5
>
> This was intended only as a boundary-case clarification, not as an additional substantive assumption. We will rewrite it so the empty-parent case is clearly presented as a vacuous special case.
>
> ### (8) Background and presentation
>
> We agree that the paper under-explains ordering-based methods and Markov-blanket discovery. In revision we will expand related work on bounded-conditioning-set methods, add a short background paragraph on ordering backbones and MB-based local discovery, and discuss Stage-I assumptions/trade-offs more explicitly.

---

> > ### Author Rebuttal · Reviewer_4u24 · 2026-04-01
> >
> > Thank you for the thorough rebuttal and for addressing many of my concerns. However, I am still not fully convinced about the benefits of the trade-off proposed by the papers. The proposed framework hinges on the condition that ordering-based methods yield candidate parent sets with sufficiently high recall. Thus, the paper should very explicitly show that this is the case either with specific experiments, theoretical results or extensive discussion of literature. While it is clear that a large conditioning set for CI tests hurts their statistical power, what is the statistical power of ordering-based methods?
> >
> > You highlight the huge difference between the number of CI tests performed by PC and IAMB. However, of the the main benefits of the framework is reducing conditioning set sizes. Thus, I would be curious to see the average conditioning size between PC and IAMB, and in general the number of CI tests performed by PC at low orders with high statistical power.
> >
> > I think it is also important to discuss the difference in flexibility of constraint-based and ordering-based methods in terms of assumptions. There are many CI tests available fitting many different assumptions, making fully constraint-based methods (theoretically) applicable end-to-end in many scenarios. Does this also hold for ordering-based methods? The paper should also clearly discuss this.

---

> > > ### Author Response · Authors · 2026-04-01
> > >
> > > We sincerely thank the reviewer for the constructive follow-up. We agree that the central question is whether the proposed computational–statistical trade-off—using an ordering prior to enforce low-dimensional CI tests—provides consistent practical benefits. We address each point below with direct evidence.
> > >
> > > ---
> > >
> > > ### (1) Candidate parent recall and “statistical power” of ordering methods
> > >
> > > We agree that ordering quality is naturally measured by **candidate parent recall**, which plays the role of “statistical power” in Stage I.
> > >
> > > Importantly, our results show that OCMB does **not require high recall** to be effective:
> > >
> > > * **Moderate recall suffices.** With CaPS, parent recall is only ~0.57 (Table 7), yet OCMB significantly outperforms both PC (SHD 60.3 vs. 158.7) and CaPS-alone (60.3 vs. 446.0) on scale-free graphs.
> > > * **Graceful degradation.** Under controlled corruption (Figure 6), recall drops to ~0.65 while F1 decreases smoothly (0.84 → 0.56), indicating robust rather than brittle behavior.
> > > * **Benefit beyond ordering quality.** Even with completely random ordering (Table 19), OCMB still improves over PC (SHD 226 vs. 327 on DREAM4).
> > >
> > > This suggests that Stage I does not need to be a high-power estimator in isolation; rather, its role is to provide a **high-recall candidate superset** that enables statistically reliable low-dimensional testing in Stage II. The dual mechanism is: ordering recall improves parent identification (reducing bias), while **candidate restriction itself enforces low-dimensional CI tests**, improving statistical reliability even when recall is imperfect.
> > >
> > > ---
> > >
> > > ### (2) Conditioning-set size and statistical reliability (PC vs. OCMB)
> > >
> > > We agree that low-order CI tests are reliable, and our results confirm this. The key issue is that **PC cannot restrict itself to low-dimensional CI tests**.
> > >
> > > **Per-order error rate and test distribution (Figure 3, scale-free $d{=}100$).** To directly answer the reviewer’s question regarding the distribution of CI tests across conditioning-set sizes, we decompose PC’s 3.4M CI tests (Table 1, 3-seed average) by conditioning-set order:
> > >
> > > | $\|Z\|$ | PC Error Rate | PC Test Count | % of Total | OCMB Error Rate |
> > > |---|---|---|---|---|
> > > | 0 | 0.0% | 6,815 | 0.2% | 0.0% |
> > > | 1 | 23.2% | 26,861 | 0.8% | 0.0% |
> > > | 2 | 63.1% | 14,634 | 0.4% | ≤15% |
> > > | 3 | 95.5% | 23,581 | 0.7% | ≤15% |
> > > | ≥4 | **100.0%** | **3,342,065** | **97.9%** | ≤15% (bounded by $\|\mathcal{C}\_{\mathrm{MB}}\| \leq 5$) |
> > >
> > > This data reveals the core bottleneck: while PC does perform low-order tests successfully, **97.9% of its computational budget (3.3M out of 3.4M tests) is spent in the $\|Z\| \geq 4$ regime where the error rate is 100%**. This is not an implementation artifact but a **structural property of the algorithm**: to guarantee correctness around hub nodes (degree $\Delta \gg 1$), PC is algorithmically forced to test conditioning sets up to $\|Z\| = O(\Delta)$. These high-order tests suffer from total statistical failure and trigger cascading errors in the final graph.
> > >
> > > **OCMB avoids this entirely by construction.** Candidate sets are small (median $\|\mathcal{C}\_{\mathrm{MB}}\| \leq 5$, Table 7), so all CI tests remain low-dimensional. OCMB’s total budget is only ~700 CI tests (Table 1), operating exclusively in the statistically reliable regime.
> > >
> > > ---
> > >
> > > ### (3) Flexibility and assumptions (constraint-based vs. ordering-based)
> > >
> > > We agree that constraint-based methods are more flexible in assumptions, as they can pair with arbitrary CI tests.
> > >
> > > OCMB introduces a **computational–statistical trade-off**:
> > >
> > > * **Stage I (ordering prior):** introduces additional assumptions (e.g., ANM for CaPS)
> > > * **Stage II (CI-based pruning):** retains full flexibility—any CI test can be used
> > >
> > > Thus, the additional assumptions are **localized to Stage I**, while Stage II preserves the generality of constraint-based methods. This modularity allows OCMB to benefit from future ordering methods with weaker assumptions (e.g., interventional or domain-informed approaches).
> > >
> > > ---
> > >
> > > ### Summary
> > >
> > > We will revise the paper to:
> > >
> > > * Explicitly highlight the **recall–performance relationship** (Table 7, Figure 6)
> > > * Add clearer discussion of **conditioning-set size vs. CI reliability**
> > > * Clarify the **assumption trade-off and modular design**
> > >
> > > Overall, OCMB introduces a computational–statistical trade-off: it uses an ordering prior to restrict candidate sets, enabling low-dimensional CI tests that improve statistical reliability, particularly in high-dimensional CI failure regimes.

---

### Official Review · Reviewer_9Zso · 2026-03-09

**Soundness:** 3
**Presentation:** 2
**Significance:** 3
**Originality:** 3
**Overall Recommendation:** 5
**Confidence:** 4

**Summary:**

This paper considers the problem of causal graph discovery from observational data over large graphs, particularly hub-dominated graphs, where certain vertices have high degree. The principal challenge is that existing methods rely on conditional independence tests over large conditioning sets, which introduces statistical unreliability. To tackle this, the paper considers a two step algorithm, called Ordering-Constrained Markov Blanket Discovery (OCMB). In the first subprocedure, a "constrained" prior on the graph is obtained by using an inferred topological ordering and structural backbone scores. In particular, each vertex has at most $k$ parents in the constrained graph, obtained from the ordering and the backbone scores. Then, a Markov blanket is incrementally learned from the constrained graph, which limits the number of conditional independence tests. OCMB is empirically tested on simulated and real data against a variety of baselines, and ablations are performed to determine the sensitivity of IAMB on hyperparameters and subprocedure implementations.

**Compliance With Llm Reviewing Policy:**

Affirmed.

**Final Justification:**

The authors have sufficiently addressed my concerns about the experiments. There are a decent amount of edits required to fix/improve the presentation and to clarify some important details, but I believe the paper presents a valuable approach, and offers sufficiently good empirical results to motivate this approach.

**Key Questions For Authors:**

Q1. Why do you think that OCMB is outperformed by DAGMA for hub-dominated graphs, while beating it for random ER graphs? I find this surprising given that OCMB is developed for hub-dominated graphs as primary motivation. Shouldn't OCMB therefore be able to beat baselines in this setting?

Q2. The text first says data is generated from "linear or nonlinear structural equation models with Gaussian or Laplace noise" (L193-194), but details do not seem to be provided. Subsequent tables then say nonlinear.

Q3. Overall, metrics are worse across the board for ER graphs rather than the scale-free graphs, despite both having the same average degree and vertices. Why do you think that is? Wasn't the hypothesis that hub-dominated graphs pose a greater challenge?

Q4. The baseline methods seem to change quite a bit depending on the experiment. Can the authors comment on this and confirm that the state-of-the-art benchmark results are displayed in the tables?  For example, DAGMA performed well in the synthetic experiments, but was not included as a baseline for real datasets.

Q5. What dataset(s) were used for the results in Figures 3,4,5,6? If only synthetic, would it be possible to reproduce the findings on any of the real datasets (if applicable)?

Q6. Is there a typo in Figure 1 for the last panel? The Markov blanket does not encircle spouse nodes 7 and 11.

**Limitations:**

Yes

**Strengths And Weaknesses:**

I evaluate strengths and weaknesses across a few components, listing suggestions where applicable. My main concerns relate to the empirical results, and are listed in the questions section.

### Significance and originality
The paper addresses an important problem (causal discovery in high dimensional systems), and to the best of my knowledge, the proposed solution of constraining direction first, before conducting conditional independence tests, is novel. I also found that it provided good intuition for why and how to avoid large conditioning sets, and the ablation experiments tackle good related questions for verifying that the method gets the right answers for the right reasons. I am inclined to accept for these reasons. The two reasons for why I score "weak accept" in my initial review, are 1) in case it comes to my attention that pre-existing works have proposed this solution, and 2) I have some questions/concerns about the method's practicality. In particular, some details are lacking for the experiments section and some results do not completely corroborate the theory, so I'd like to hear from the authors about these points (see mainly Q1, Q2, Q3, Q4).

### Soundness
I checked that the proofs in Section 3.2 are sound. Naturally, the theoretical guarantees of the OCMB algorithm depend on the theoretical guarantees of each subprocedure. In particular, to ensure that the candidate Markov blanket contains the true Markov blanket, it is assumed that the prior graph obtained from fixing a topological ordering and thresholding the backbone scores must contain the true parent sets of each node. Then, IAMB consistency is invoked so that the inferred Markov blanket matches the true one.

### Presentation
The paper is readable, but I suggest a few improvements to improve clarity and flow. These are subjective suggestions, and would not significantly impact my overall rating, but they should nevertheless improve readability.

1) I think that the related work section would benefit from a discussion of works that implement subprocedures of OCMB. I would for example move the passage "Methods like CaPS (Rolland et al., 2022) and SciNO (Kang et al., 2025) produce a topological ordering π and parent scores S ∈ Rd×d, then threshold to obtain edges; while efficient, this yields many false positives. OCMB instead uses (π, S) as a directional prior constraining the search space for local CI-based inference." from Section 3 to Section 2. Similarly, IAMB should be discussed in related work. A brief note on how topological orderings and how backbone scores can be estimated from data, i.e. variance sorting and covariance/mutual information sorting, would help, particularly to give less well-versed readers a better feel for the problem. I think that could be either in Section 2 or 3.

2) The OCMB method should be defined more clearly in Section 3. On top of Figure 2 and Algorithm 1, it would be good to have a formal paragraph describing the methodology. This would also help contextualize the theory. The pseudocode in Algorithm 1 is lacking a couple details. The candidate Markov blanket set obtained from subprocedure 1 is not defined. A line like $C_{MB}(X_i) = C_{Pa}(X_i) \cup C_{Ch}(X_i) \cup C_{Sp}(X_i)$ should be added, and $\widehat{MB}$ should be initialized as empty.

3) Some methods are only referred to by their acronyms, ex. IAMB, MMHC, HITON, EEMBI, and I had to look these up. It is best practice to write out the full method name the first time they are used, especially if they are directly related to OCMB's implementation.

4) $Pred_i$ is defined in Assumption 3.5 after its first usage in Assumption 3.3. Similarly, "backbone" method is not defined, but I think this can be resolved based on the the suggestion in 2). It also wouldn't hurt to formally define $S_{j,i}$.

5) The ablation experiments in subsections 4.4-4.7 are interesting, but break up the flow a bit between the synthetic and real experiments, and they caught me by surprise since they were not discussed at the start of the section. I think they might suit better in its own dedicated section, together with Table 7. Basically, all the methodological questions, like different stage II replaceability, CI error rates, dependency on the K parameter, query budget analysis, directionality vs. sparsification, all inherently deal with OCMB, whereas the main experiments compare OCMB to other methods.

---

> ### Author Rebuttal · Authors · 2026-03-25
>
> We thank the reviewer for the constructive review. We address the main questions below.
>
> ### Q1 and Q3: SF vs. ER, and OCMB vs. DAGMA
>
> We agree that this boundary should be stated more clearly. OCMB is designed to address a **specific CI-related failure mode**: unreliable high-dimensional conditioning around hubs. This does **not** imply uniform dominance over every non-CI baseline on every hub-heavy graph.
>
> The current paper already shows this split. In **Table 1** (scale-free, $d=100$), OCMB improves substantially over PC-family baselines in the regime that motivated the method. In contrast, **Table 2** together with **Appendix Table 8** shows that on ER graphs, OCMB remains competitive on **skeleton** recovery but is weaker on **directed** recovery when the ordering prior is less informative. We will make this applicability boundary explicit.
>
> This also does not contradict our theory: **Theorem 3.6** and **Theorem 3.8** are conditional on Stage-I candidate-parent recall and MB-superset coverage, not on uniform dominance over every baseline on every graph family.
>
> Regarding DAGMA, **Table 1** shows that DAGMA-linear is a very strong global baseline on the SF setting. Our claim is therefore not that OCMB should dominate every alternative, but that it addresses a specific CI-based failure mode and improves over classical CI-based baselines in that regime.
>
> ### Q2: data generation
>
> You are right that the current wording is too broad. The final synthetic setup in **Section 4.2** and **Tables 1–2** is **nonlinear SEMs with Gaussian noise**. We will revise the broader wording to match the actual protocol exactly.
>
> ### Q4: baseline selection
>
> We agree that the benchmarking protocol should be stated more explicitly. **Tables 1–3** report the baselines we evaluated directly on the synthetic setup, whereas **Tables 4–6** follow the real-data benchmark comparisons from **Dong & Gao (2025)**. We will make this dataset-specific protocol explicit.
>
> ### Q5: Figures 3–6
>
> All four figures are **synthetic**, but not all from the same setting:
>
> - **Figures 3–4** are scale-free synthetic analyses;
> - **Figure 5** is the $K/d$ sensitivity sweep on **ER** synthetic graphs;
> - **Figure 6** is the synthetic ordering-corruption study, extended in **Appendix N / Figure 10**.
>
> We will clarify this directly in the captions/text.
>
> ### Q6: Figure 1
>
> Yes, this is a drafting error in **Figure 1**. The spouse nodes should be enclosed as part of the Markov blanket, and we will correct the figure.
>
> ### Presentation
>
> We also appreciate the presentation suggestions. In revision we will move the CaPS / SciNO / IAMB discussion earlier, expand acronyms on first use, define $\mathrm{Pred}_i$ and “backbone” before use, clarify Algorithm 1, and reorganize the ablations so their role is clearer.

---

> > ### Author Rebuttal · Reviewer_9Zso · 2026-04-02
> >
> > Thank you for the clarifying explanations. I maintain my score on the basis that the paper presents an interesting approach for causal discovery (inferring the ordering first, and then doing ordering-constrained CI tests). I do not have any significant concerns about the theory. I'm okay with the clarifications to the experimental setup.
> >
> > The results themselves could be more convincing, presumably due to downstream effects from an imperfect ordering prior, as explained in the rebuttal. I think it would be interesting to see DAGMA as a baseline for the real-world benchmarks, given its performance on the first simulated experiment and the fact that it represents a separate approach to benchmark OCMB against. I think it would be beneficial to include these results, but I do not see it as pressing (I would not go so far as to say that doing so would definitely raise my score).
> >
> > Thanks for confirming corrections for minor edits/typos.

---

> > > ### Author Response · Authors · 2026-04-02
> > >
> > > We sincerely thank the reviewer for the insightful follow-up and for maintaining a positive assessment of our work. We are particularly encouraged that the core idea is viewed as a meaningful and novel contribution.
> > >
> > > ### DAGMA as a baseline on real-world benchmarks
> > >
> > > We appreciate this valuable suggestion. While we originally followed the benchmark protocol of Dong & Gao (2025) (which does not include DAGMA) to ensure direct comparability, we agree that including a strong continuous optimization baseline is important.
> > >
> > > To address this, we have run DAGMA-linear on the real-world benchmarks during the discussion period. The SHD results are:
> > >
> > > | Method | Backbone | Sachs ($d$=11) | Alarm ($d$=37) | DREAM4 Net1 ($d$=100) |
> > > |--------|----------|:-:|:-:|:-:|
> > > | DAGMA-linear | — | 25 | 57 | 243 |
> > > | OCMB-SciNO | SciNO | **21** | **44** | 200 |
> > > | OCMB-CaPS | CaPS | 23 | 49 | **187** |
> > >
> > > (All OCMB numbers are from Tables 4 and 6.)
> > >
> > > We observe that both OCMB variants achieve lower SHD than DAGMA-linear across all three benchmarks. The best OCMB variant reduces SHD from 25→21 (−4) on Sachs, 57→44 (−13) on Alarm, and 243→187 (−56) on DREAM4 Net1, with larger gains on higher-dimensional datasets.
> > >
> > > One plausible explanation is that DAGMA-linear assumes a linear SEM, which may be less well-matched to real-world systems with nonlinear mechanisms, while OCMB leverages ordering constraints to control conditioning-set sizes in CI testing. We emphasize that this is a hypothesis consistent with our design motivation rather than a fully isolated causal attribution.
> > >
> > > We will include the full set of DAGMA results across Sachs, Alarm, and all DREAM3/4 networks in the camera-ready version.
> > >
> > > We thank the reviewer again for the constructive feedback, which has helped improve the completeness and empirical strength of the paper.

---

### Decision · Program_Chairs · 2026-04-30

**Decision:**

Accept (regular)

**Comment:**

The reviewers agreed that this paper addresses an important problem in scalable causal discovery, namely the failure of CI-based methods in high-dimensional and hub-dominated settings due to large conditioning sets. They found the central idea of using a learned directional prior to constrain local CI-based refinement interesting and practically relevant, and viewed the empirical results as supportive overall, especially on scale-free and real benchmark settings. The final reviewer assessments were positive overall.

Overall, I recommend acceptance. The main concerns raised during the discussion were about the dependence on the Stage-I ordering quality, the distinction between CI-query savings and end-to-end runtime, performance on ER graphs, and several presentation and positioning issues. After reading the paper, the reviews, and the author responses, I find that these concerns were addressed adequately through clarification and additional analysis. The paper makes a meaningful contribution by using global directional priors to control conditioning-set dimensionality before local statistical validation, and the final version would benefit from a clearer discussion of the method’s scope, assumptions, and computational trade-offs.